# Do 3D Large Language Models Really Understand 3D Spatial Relationships?

**Xianzheng Ma[1,2]*  Tao Sun[3]*  Shuai Chen[1,2]  Yash Bhalgat[1]  Jindong Gu[5] †**
**Angel X Chang[4]  Iro Armeni[3]  Iro Laina[1]  Songyou Peng[5] ‡  Victor Adrian Prisacariu[2] ‡**
[1] VGG, University of Oxford  [2] AVL, University of Oxford  [3] Stanford University
[4] Simon Fraser University  [5] Google DeepMind
Code & Dataset: https://real-3dqa.github.io/

## ABSTRACT

Recent 3D Large-Language Models (3D-LLMs) claim to understand 3D worlds, especially spatial relationships among objects. Yet, we find that simply fine-tuning a language model on text-only question-answer pairs can perform comparably or even surpass these methods on the SQA3D benchmark without using any 3D input. This indicates that the SQA3D benchmark may not be able to detect if the model exploits textual shortcuts rather than engages in 3D-aware reasoning. To address this issue, we introduce **Real-3DQA**, a more rigorous evaluation benchmark that filters out easy-to-guess questions and introduces a structured taxonomy to assess various aspects of 3D reasoning. Experiments on Real-3DQA confirm that existing 3D-LLMs struggle with spatial relationships once simple cues are removed. We further propose a **3D-reweighted training objective** that guides model to rely more on 3D visual clues, substantially enhancing 3D-LLMs' performance in spatial reasoning tasks. Our findings underscore the need for robust benchmarks and tailored training strategies to advance genuine 3D vision-language understanding.

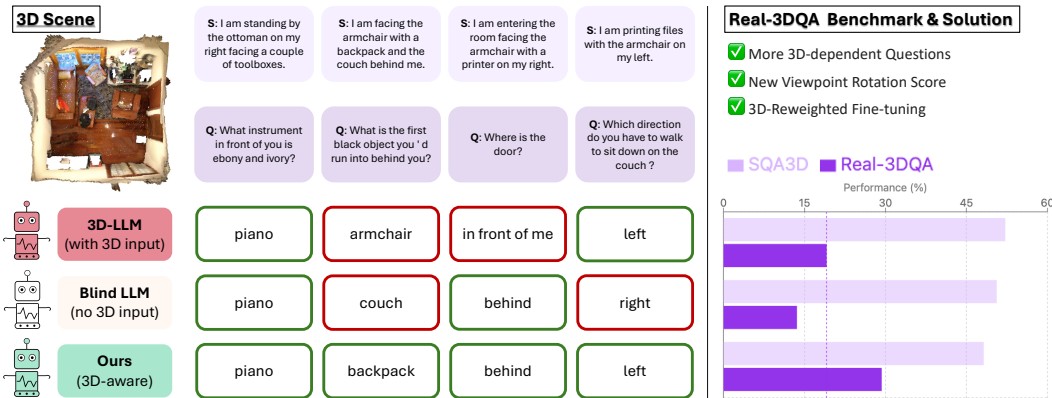

**Figure 1: Our Real-3DQA Benchmark and Solution.** We demonstrate the answer differences between 3D-LLM (LEO) and its blind-finetuned version on various questions, with correct answers highlighted in green frames and incorrect ones in red frames. We discover that some questions can be answered correctly regardless of whether 3D information is used, which we consider to be 3D-independent questions. By filtering out these 3D-independent questions and introducing a new viewpoint rotation score, the original models' performance drops significantly on Real-3DQA. Finally, using our proposed 3D-aware Reweighted Finetuning strategy, performance improves again.

---

*Equal contribution
†Correspondence author
‡Equal supervision

# 1 INTRODUCTION

Understanding and reasoning about 3D environments is fundamental for embodied AI, AR/VR applications, autonomous driving, etc. Recent work has proposed 3D Large Language Models (3D-LLMs) Hong et al. (2023); Chen et al. (2024); Guo et al. (2023); Huang et al. (2024a), aiming to empower language models with spatial awareness. Early evaluations focused on tasks such as 3D captioning Chen et al. (2020) and object grounding Yang et al. (2024); Achlioptas et al. (2020), while 3D question answering (3D-QA) Azuma et al. (2022); Ye et al. (2021); Etesam et al. (2022) soon emerged as a more flexible benchmark for assessing diverse aspects of spatial understanding. More recently, 3D Situated QA Ma et al. (2023); Zhang et al. (2025c); Linghu et al. (2025) extended this line of evaluation by adopting an egocentric, first-person perspective, simulating how humans perceive the world and challenging models to reason continuously about objects and relations in context. While these benchmarks are intended to measure progress toward robust embodied intelligence, we question whether the reported improvements truly reflect genuine 3D spatial reasoning. This paper investigates this gap between perceived progress and actual spatial understanding.

We find that much of this apparent progress is illusory. On the most-widely used situated QA benchmarks, i.e. SQA3D Ma et al. (2023), where accuracy has increased from 30% to over 50% in recent years, 3D-LLMs can achieve competitive scores without actually processing any 3D data. A simply fine-tuned *blind* model[1] finetuned on text-only question-answer pairs can match or surpass the performance of recent 3D-LLMs, as illustrated in Fig. 2. This reveals that many SQA3D questions can be solved solely by linguistic priors, rather than genuine spatial reasoning. Importantly, we observe similar vulnerabilities beyond SQA3D, including on ScanQA and the more recent MSR3D benchmark. Meanwhile, our findings echo those of Huang et al. (2025); Li et al. (2025); Zhang et al. (2024); Li et al. (2025) and indicate that even the latest 3D-LLMs are not immune to this gap.

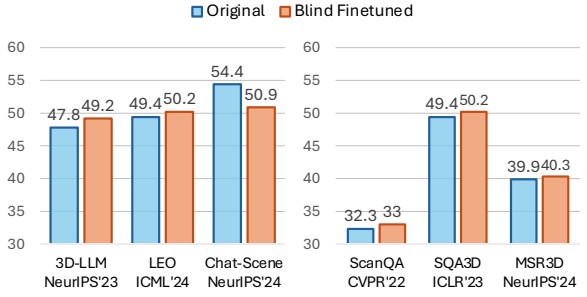

Figure 2: **Our finding**: A language model fine-tuned only on text QA pairs without any 3D inputs (Blind Finetuned) can match or even surpass state-of-the-art 3D-LLMs (Original) on multiple 3D-QA benchmarks. This exposes a critical weakness in current benchmark design and calls into question their ability to assess genuine 3D reasoning despite linguistic shortcuts.

What causes existing benchmarks to fall short in accurately evaluating 3D reasoning? One hypothesis is the presence of bias Vo et al. (2025) in 3D-QA datasets. These biases can stem from both manual annotation Azuma et al. (2022); Ye et al. (2021); Zhang et al. (2025b); Huang et al. (2025) and automatic or semi-automatic LLM-assisted generation processes Linghu et al. (2025); Zhang et al. (2025c), where linguistic or commonsense priors are embedded in the data. For example, "What is the black rectangular object on the wall?" almost always yields "TV" for indoor scenes regardless of spatial context. These questions allow models to succeed via shortcut learning instead of true 3D understanding. While SQA3D Ma et al. (2023) attempt to mitigate this issue by balancing answer distributions, such surface-level fixes cannot eliminate deeper biases, like preferences for salient objects, canonical configurations, or easily guessable answers. These dataset biases are multi-faceted and difficult to eliminate through either manual correction or distribution balancing.

To address this, we introduce Real-3DQA, a benchmark that filters out questions solvable by language priors alone, as summarized in Table 1. We compare each 3D-LLM's accuracy to its "blind" counterpart—finetuned on text-only QA pairs without any 3D inputs. Questions answered correctly by both models are considered low in *3D dependency* and are removed, as the answer can be inferred without spatial understanding. By filtering out such trivial questions, we implicitly mitigate a broad range of biases without requiring detailed manual intervention. This model-based filtering allows our Real-3DQA more robustly evaluate genuine 3D reasoning.

---

[1] We refer readers to section 4.1 for details on how the *blind* finetuned model is obtained.

Although removing low–3D–dependency questions prevents linguistic shortcuts, it does not guarantee that a model truly understands spatial structure. To further assess genuine 3D understanding, we propose a *cross-question consistency* test under viewpoint changes. The key idea is that if a model truly grasps the 3D scene, it should answer the same question correctly even when the observer's orientation changes. Concretely, we rotate the viewpoint while keeping the position and question fixed, and adjust any directional terms in both the situation description and the expected answer. This results in logically equivalent QA pairs across different reference frames, forcing the model to interpret spatial relations consistently. To quantify this, we further introduce the Viewpoint-Rotation Score, which measures whether the model maintains spatial consistency across varied perspectives.

While our proposed benchmark and consistency test offer a more rigorous evaluation, they do not by themselves prevent models from learning shortcuts during training. We find that standard supervised fine-tuning still encourages models to learn linguistic shortcuts, limiting the benefit of 3D information. To address this limitation from the training perspective, we introduce a 3D-aware Reweighted Fine-tuning (3DR-FT) strategy. Our key idea is to quantify the 3D dependency of each question by measuring the performance gap between a blind (text-only) model versus a full 3D-LLM. Based on this gap, we downweight questions that can be guessed easily from text alone, and upweight those requiring genuine spatial reasoning. This adaptive reweighting encourages models to incorporate 3D cues, and experiments confirm consistent performance gains across multiple 3D-LLMs on our revised benchmark.

Overall, our contributions can be summarized as below:

- We introduce a new diagnostic benchmark to **measure genuine 3D reasoning** ability of 3D-LLMs. It has filtered out questions solvable by linguistic shortcuts and uses a viewpoint-rotation score to provide a stricter and more reliable evaluation.

- We reveal that **modern 3D-LLMs are surprisingly brittle**. On our benchmark, their performance drops by over 60%, and they almost completely fail our viewpoint consistency tests, a critical flaw overlooked by prior benchmarks.

- To fix this, we propose a **3D-aware reweighted fine-tuning** strategy that forces models to focus more on spatially complex data, consistently boosting the 3D reasoning capabilities of LLMs.

## 2 RELATED WORK

**3D-LLMs.** 3D-LLMs integrate spatial data with LLMs for scene understanding Ma et al. (2024b), including 3D grounding, captioning, and question answering (QA). Prevalent approaches typically follow a formulation similar to 2D vision-language models (VLMs), by concatenating 3D tokens from point-cloud encoders with text tokens before being processed by an LLM Hong et al. (2023); Yang et al. (2023); Wang et al. (2023); Huang et al. (2023); Fu et al. (2024); Xu et al. (2024); Yang et al. (2024); Chen et al. (2024); Li et al. (2024); Huang et al. (2024b); Guo et al. (2023). Others incorporate additional visual signals, such as multi-view images, into point-cloud-based 3D-LLMs. This hybrid design, exemplified by LEO Huang et al. (2024b) and Chat-Scene Huang et al. (2024a), has been shown to improve 3D captioning, grounding, QA, and Situated QA (SQA). More recently, methods such as GPT4Scene Qi et al. (2025), LLaVA-3D Zhu et al. (2024), and Video-3D LLM Zheng et al. (2024) sidestep explicit 3D encoders and instead rely on 2D VLMs for 3D awareness. While these works primarily aim to boost model accuracy through architectural and training innovations, we take a different perspective: questioning whether current 3D-LLMs truly acquire consistent understanding of 3D spaces. Our findings suggest that the answer is far from affirmative.

**Linguistic Biases in Multi-modality Models.** The issues related to the exploitation of linguistic bias have been extensively studied in 2D-VLMs Agrawal et al. (2018); Niu et al. (2021); Ouyang et al. (2021); Ma et al. (2024a), leading to fairer benchmarks Agrawal et al. (2018) and mitigation strategies Ma et al. (2024a), such as ensemble learning, data augmentation, self-supervised contrastive learning, and response re-ranking. However, the challenges of mitigating similar exploitation in 3D-QA benchmarks remain largely unexplored. Many existing 3D-QA datasets rely on automatic or semi-automatically generated QA pairs Yan et al. (2023); Li et al. (2023); Qian et al. (2024), which often include language priors in their questions. Although Ma et al. (2023) balances answer categories, it cannot remove deeper annotation artifacts (e.g., saliency preferences, canonical layouts,

Table 1: **Comparison of 3D Question Answering Benchmarks.** Our Real-3DQA benchmark features situated questions, uses LLM-assisted text collection with human verification, applies debiasing techniques, evaluates robustness across different viewpoints, and provides diverse question categories to assess true 3D spatial reasoning ability.

| Dataset | Situated | Text collection | Debiased | Robustness evaluation | Quality check | Question categories |
|---|---|---|---|---|---|---|
| ScanQA | ✗ | Human | ✗ | ✗ | ✗ | 0 |
| 3D-QA | ✗ | Human | ✗ | ✗ | ✗ | 4 |
| SQA3D | ✓ | Human | ✓ | ✗ | ✓ | 5 |
| MSQA | ✓ | LLM | ✗ | ✗ | ✓ | 6 |
| ScanReQA | ✗ | Human | ✗ | ✓ | ✓ | 0 |
| Spartun3D | ✓ | LLM | ✗ | ✗ | ✓ | 3 |
| Beacon3D | ✗ | Human | ✗ | ✗ | ✓ | 5 |
| **Real-3DQA** | ✓ | LLM | ✓ | ✓ | ✓ | **11** |

or guessable questions). We instead reduce linguistic bias via *model-level* comparison: by contrasting a full model with its *blind* (text-only) fine-tuned variant, we filter items that are solvable through language priors alone. This consistency-based procedure offers a more robust alternative to purely rule-based or distributional balancing.

**3D-QA Diagnostic Benchmarks.** Existing 3D-QA benchmarks largely reuse the 2D-QA protocol of per-item accuracy Rajpurkar et al. (2016), which overlooks whether predictions are self-consistent in 3D spaces. To overcome such issues, the recent diagnostic benchmark Beacon3D Huang et al. (2025), instead assesses 3D understanding through cross-task consistency: whether the model gives consistent answers across both QA and grounding tasks. In contrast, we evaluate 3D understanding ability through cross-question consistency, where we test if the model consistently solves the question under different viewpoint augmentations.

## 3 REAL-3DQA BENCHMARK DESIGN

Situated Question Answering (SQA) challenges AI to comprehend its position and surroundings within an environment before responding to queries. The first benchmark for this task, SQA3D Ma et al. (2023), has been widely adopted by recent research to evaluate the spatial awareness and 3D reasoning capabilities of 3D-LLMs. However, as shown earlier in Figure 2, we discovered a critical flaw in the current evaluation benchmark, which reveals that current 3D-LLMs obtain comparable or even worse results than its blind counterpart trained with text input only.

This observation underscores the need for a more rigorous evaluation framework. To address this gap, we introduce Real-3DQA, a benchmark derived from SQA3D to more accurately assess 3D spatial reasoning in 3D-LLMs. The key innovations in constructing Real-3DQA consist of two main steps: (1) **Filtering 3D-independent Questions** ( Section 3.1), we identify and filter out those questions that can be answered by text context alone with high probability, minimizing the model's reliance on dataset biases and ensuring a more genuine assessment of 3D reasoning capabilities. (2) **Viewpoint Rotation Score** ( Section 3.2), we introduce a viewpoint rotation evaluation metric that complements existing SQA3D metrics, encouraging models to rely on genuine 3D understanding.

### 3.1 FILTERING 3D-INDEPENDENT QUESTIONS

In this section, we describe how we remove questions that are "easy-to-guess" without 3D spatial information. Let $Q$ be the original set of test questions from Ma et al. (2023). We evaluate three different 3D-LLMs Huang et al. (2024b;a); Hong et al. (2023), denoted as $M_A$, $M_B$, and $M_C$. For each specific model $X$, we begin with training the original model $M_X$, where $X \in \{A, B, C\}$ and its *blind* counterpart $M_X^{\text{blind}}$, which is the same model finetuned without 3D input[2].

Next, we utilize both $M_X$ and $M_X^{\text{blind}}$ models to identify questions that are *independent of 3D input*. Based on our definition, a question $q \in Q$ is considered *independent of 3D input* if both the original and *blind-finetuned* models predict the correct answer: $M_X(q) = M_X^{\text{blind}}(q) =$ correct. For any given model $M_X$. The set of such questions for each model is: $Q_X = \{q \in Q \mid M_X(q) = M_X^{\text{blind}}(q) =$

---

[2] We refer readers to Section 4.1 for details on how Blind Finetuned model $M_X^{\text{blind}}$ is obtained

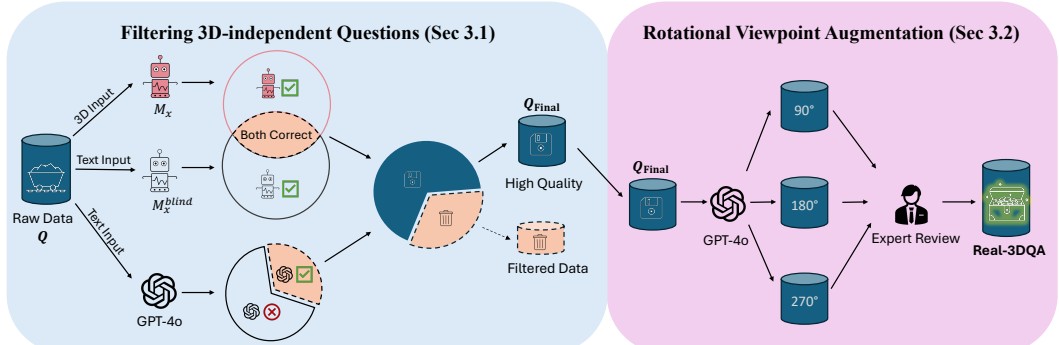

Figure 3: **Overview of Real-3DQA Construction Process.** Real-3DQA provides a fair and rigorous evaluation framework for 3D spatial reasoning in 3D-LLMs. The construction process begins with Filtering 3D-independent Questions, which removes questions that can be correctly answered by both the 3D-LLM model $M_x$ and its text-only $M_x^{blind}$ counterpart, as well as those answerable by the GPT model without 3D input. The remaining high-quality questions $Q_{Final}$ are then augmented using GPT, generating spatially consistent variations through viewpoint rotations while preserving the underlying 3D relationships. Finally, expert reviews eliminate redundancy and invalid data, ensuring the highest dataset quality.

correct}. To ensure robustness, we repeat this for all three models and take the union of questions that are *independent of 3D input*: $Q_{\text{3D-filtered}} = Q_A \cup Q_B \cup Q_C$. The underlying hypothesis for this approach is that if a set of questions can be answered correctly by both the original and *blind-finetuned* models, it means the question set is both less relevant to 3D spatial understanding and can be overfitted by data priors. Thus, we obtain a more rigorous remaining question set by removing those 3D-independent and trivial questions: $Q' = Q \setminus Q_{\text{3D-filtered}}$. We refer the reader to Figure 3 to illustrate the procedure up to this stage.

To ensure the final benchmark exclusively assesses true 3D understanding, we further refine $Q'$ by removing questions that a general large language model (*i.e.*, GPT-4o-mini OpenAI (2024)) can correctly answer based solely on the textual content of the *situation* and *question* without 3D input. Assuming the set of questions, GPT can answer using only textual input as: $Q_{\text{GPT}} = \{q \in Q' \mid \text{GPT}(q) = \text{correct}\}$. The final step to obtain the filtered test set is then: $Q_{\text{final}} = Q' \setminus Q_{\text{GPT}}$. In Section D.1 of Appendix, we present the change in the number of questions at each step of the filtering (Table 7). We also provide illustrative examples of 3D-independent questions that were removed.

## 3.2 VIEWPOINT ROTATION SCORE

To comprehensively assess the effectiveness of the Real-3DQA test set, we employ a combination of standard evaluation metrics and a viewpoint rotation metric specifically designed to measure true 3D understanding. First, we adopt the same evaluation metrics used in SQA3D, such as refined exact match (EM_R) Huang et al. (2024b), ensuring comparability with existing benchmarks. For a detailed analysis of their impact on model performance, we refer the readers to Section 5, which presents comprehensive experimental results.

To reduce the influence of superficial textual cues on model performance, we propose the Viewpoint Rotation Score (VRS) to assess whether 3D-LLMs genuinely comprehend spatial relationships. The core idea is to generate rephrased test questions by rotating the situation descriptions and corresponding answers from the original seed question while preserving underlying 3D relationships. Figure 4 illustrates this process: in the original scenario, an agent faces a trash can with a table behind and asks,"What is on my right?" The correct answer is "whiteboard". By rotating the agent's viewpoint (e.g., 90°, 180°, 270°), we create alternative descriptions where the agent faces a different object, yet spatial relationships of scene objects remain unchanged. The ground-truth answer dynamically updates based on the new viewpoint. We employ GPT to generate augmented SQA questions by processing pre-designed templates and scene graphs from Ma et al. (2023) from the original question. We provide our prompt template in Figure 9 of the Appendix D.

Figure 4: **Viewpoint Rotation Augmentation.** Real-3DQA generates viewpoint-augmented SQA instances to enforce spatial reasoning. The left panel shows the original room layout, where an agent asks "What is on my right?" with the correct answer "white board." The right panels illustrate the SQA examples of the original viewpoint and rotated viewpoint (90°, 180°, 270°), where the agent's perspective shifts and the correct answers dynamically adjust while preserving spatial consistence .

**Quality Control.** To ensure the reliability of our augmented dataset, we adopt a multi-stage quality control process combining GPT hallucination mitigation and structured expert review. We guide GPT generation with in-context examples, apply manual validation to enforce rotation sensitivity and QA validity, and conduct expert reviews with qualification checks and agreement metrics. We also use multiple vision-language-models to cross-validate the augmented QA pairs, confirming that they are faithful to 3D rotations and grounded in genuine spatial reasoning. Full details of this process are provided in Appendix D.2.

**VRS evaluation metric design.** VRS measures the robustness of a model to spatial variations on a scale of 0 to 100%, where higher values indicate better performance. During the evaluation, each batch consists of four related questions: the original and its three rotated variants (90°, 180° and 270°). After processing the entire data set, we calculate the percentage of instances in which the model correctly answers at least $k$ questions, where $k \in \{1, 2, 3, 4\}$. Formally, let $N_k$ be the number of instances where at least $k$ questions in a batch are answered correctly, and $N_{\text{total}}$ be the total number of instances. The percentage for each $k$ is given by: $P_k = \frac{N_k}{N_{\text{total}}} \times 100$ ,where $P_k$ represents the accuracy percentage for answering at least $k$ questions correctly. The final VRS metric is computed as the mean of these percentages: $\text{VRS} = \frac{1}{4} \sum_{k=1}^{4} P_k$.

To achieve high performance in VRS, the 3D-LLM are expected to correctly answer all variations of the same question, demonstrating a true understanding of 3D spatial relationships rather than relying on textual patterns. Intuitively, responses inconsistently across rephrased questions reveal weaknesses in the model's 3D reasoning capabilities. By design, VRS is a stricter evaluation metric than the standard ones mentioned previously, as it penalizes models that fail to provide consistent answers across multiple question variants. In addition, taking the mean of $P_k$ ensures that the metric does not disproportionately favor cases where the model succeeds on only a subset of viewpoints.

## 4 3D-AWARE FINE-TUNING STRATEGY

### 4.1 ORIGINAL FINE-TUNING

Let the 3D-QA dataset $\mathcal{D}$ consist of triplets $(\boldsymbol{x}_{\text{text}}^{(i)}, \boldsymbol{x}_{\text{3D}}^{(i)}, \boldsymbol{y}^{(i)}) \sim \mathcal{D}$, where $\boldsymbol{x}_{\text{text}}$ represents a textual prompt that includes a situational description and a corresponding question, $\boldsymbol{x}_{\text{3D}}$ denotes the associated 3D context (*e.g.*, point clouds), and $\boldsymbol{y}$ is the ground truth answer.

**Supervised Fine-tuning.** In a typical 3D-LLM *Supervised Fine-Tuning* (SFT) setting Huang et al. (2024b); Wang et al. (2023); Huang et al. (2023); Chen et al. (2024), we fine-tune the model on both text and 3D data to learn mappings from 3D representations to the correct answers. However, the SFT process doesn't consider how the model uses the data. Empirically, we observe that models still tend to learn over-rely on textual cues rather than learning from 3D context, what we refer to as a textual shortcut problem.

**Blind Fine-tuning.** To highlight this potential textual shortcut issue, we next consider *Blind Fine-tuning* (BF), where we completely ignore the 3D context $\boldsymbol{x}_{\text{3D}}$ and train the model only on textual inputs. Formally, we fine-tune the model parameters $\theta$ using the standard next-token cross-entropy loss, but conditioned only on the textual prompt $\boldsymbol{x}_{\text{text}}$. Using BF, we optimize a blind model, which reveals how much of the 3D question-answering task can be solved (or guessed) by language context alone. Furthermore, we leverage the BF model as a reference for assessing the true benefit of 3D data.

## 4.2 3D-AWARE REWEIGHTED FINE-TUNING

Motivated by the observation of SFT and BF models, we propose 3D Reweighted Fine-tuning (3DR-FT). Our goal is to explicitly encourage the model to use the 3D context, rather than "shortcutting" through language patterns. We begin with the BF model $p_\phi$ and upweight samples adaptively for which the BF model finds more *difficult to guess* compared with current model, ensuring that the final model is pushed to 3D dependency.

**Reweighting function.** We define a reweighting function $w_j(\boldsymbol{y}, \boldsymbol{x}_{\text{text}})$ based on the ratio of *surprise*. It measures how surprised the blind model $p_\phi$ is when it finds out about the ground truth token $\boldsymbol{y}_j$, compared to the current training model $p_\theta$. More formally, we define

$$w_j(\boldsymbol{y}, \boldsymbol{x}_{\text{text}}) := \frac{S_\phi(\boldsymbol{y}, \boldsymbol{x}_{\text{text}})}{S_\theta(\boldsymbol{y}, \boldsymbol{x}_{\text{text}})} = \frac{\log p_\phi(\boldsymbol{y}_j \mid \boldsymbol{y}_{<j}, \boldsymbol{x}_{\text{text}})}{\log p_\theta(\boldsymbol{y}_j \mid \boldsymbol{y}_{<j}, \boldsymbol{x}_{\text{text}})}, \tag{1}$$

where $S_\theta(\boldsymbol{y}, \boldsymbol{x})$ is the surprise function Ash (2012) of predicting the last token in $\boldsymbol{y}$ given $\boldsymbol{x}$. Higher values of $w_j$ indicate that the BF model places a higher surprise on ground truth token $\boldsymbol{y}_j$ than the current model, indicating the token is harder to guess by textual context alone. Naturally, we increase the weights to those tokens in the loss function. We then define the 3DR-FT loss function as:

$$\mathcal{L}_{\text{3DR-FT}}(\theta) := \mathbb{E}_{\mathcal{D}}\Big[-\sum_{j=1}^{T} w_j(\boldsymbol{y}, \boldsymbol{x}_{\text{text}}) \log p_\theta(\boldsymbol{y}_j \mid \boldsymbol{y}_{<j}, \boldsymbol{x}_{\text{text}}, \boldsymbol{x}_{\text{3D}})\Big]. \tag{2}$$

In Appendix E, we provide theoretical insights on how this objective can promote the model's 3D-context dependency.

## 5 EXPERIMENTS AND ANALYSIS

Our experiments try to answer the following questions: (1) How does our benchmark differ from SQA3D in evaluating true 3D understanding? (2) Do existing 3D-LLMs exhibit rotation robustness when evaluated with our proposed *Viewpoint-Rotation Score*? (3) Does our 3D-Reweighted Fine-tuning strategy enhance 3D dependency and improve performance on Real-3DQA?

### 5.1 EXPERIMENTAL SETUP

We evaluate five representative 3D-LLM models (Hong et al., 2023; Huang et al., 2024b;a; Qi et al., 2025; Huang et al., 2023) across ten reasoning abilities on both the SQA3D and Real-3DQA benchmarks. For each model, we test their released pretrained model weights. To demonstrate the effectiveness of our training strategies, we select more advanced models LEO and Chat-Scene as baselines, then apply our proposed training strategy.

### 5.2 BENCHMARK EFFECTIVENESS

**Performance comparison with SQA3D.** To assess the increased difficulty of Real-3DQA, we compare model performance before and after the benchmark update. As shown in Tab. 2, the updated benchmark leads to a substantial drop in the Exact Match (EM) metric: 40.3 for 3D-LLM Hong et al. (2023), 41.6 for Chat-3D v2 Huang et al. (2023), 35.1 for LEO Huang et al. (2024b), 37.4 for Chat-Scene Huang et al. (2024a), and 27.5 for GPT4Scene Qi et al. (2025), relative to the SQA3D benchmark. Refined EM metrics show similar declines. These results illustrate that our benchmark

Table 2: **Performance comparison on SQA3D and our Real-3DQA**. We see a significant performance drop in our new benchmark.

| 3D-LLMs | Venue | SQA3D | | Real-3DQA | |
|---|---|---|---|---|---|
| | | EM | EM_R | EM | EM_R |
| 3D-LLM | NeurIPS 23 | 47.8 | 49.6 | 7.5 | 10.4 |
| Chat-3D v2 | Arxiv 24 | 45.0 | 48.1 | 3.4 | 9.7 |
| LEO | ICML 24 | 49.4 | 52.2 | 14.3 | 19.1 |
| Chat-Scene | NeurIPS 24 | 54.4 | 57.2 | 17.0 | 22.1 |
| GPT4Scene | ICLR 26 | **60.6** | **63.3** | **33.1** | **36.9** |

poses greater challenges and better exposes differences in 3D understanding, as methods with similar SQA3D performance (e.g., 3D-LLM, Chat-3D v2, and LEO) diverge more clearly here.

To verify the effectiveness of our proposed viewpoint rotation evaluation, we conduct a probing experiment on five 3D-LLMs; results of the correct times under four rotations, the overall VRS, and

Table 3: **Rotation Robustness Comparison on Real-3DQA.** The table shows the performance of different 3D-LLMs (using refined exact match metric) when tested with varying numbers of correct rotations. All models demonstrate a clear performance degradation as the required number of correct rotations increases, highlighting the challenge of rotation robustness in 3D understanding.

| 3D-LLMs | Correct times | | | | VRS | Question Types | | | |
|---|---|---|---|---|---|---|---|---|---|
| | one | two | three | four | % | distance | direction | counting | existence |
| 3D-LLM | 33.2 | 4.1 | 1.1 | 0.1 | 9.6 | 5.5 | 24.6 | 24.2 | **44.2** |
| Chat-3D v2 | 23.2 | 2.7 | 0.5 | 0.0 | 6.6 | 4.3 | 14.6 | 16.7 | 21.2 |
| LEO | 46.9 | 8.1 | 1.6 | 0.4 | 14.3 | 11.8 | 23.2 | 20.0 | 36.5 |
| Chat-Scene | 43.3 | 7.1 | 1.2 | 0.1 | 12.9 | 10.1 | 23.7 | 16.7 | 40.4 |
| GPT4Scene | **55.5** | **14.3** | **2.5** | **0.5** | **18.2** | **15.8** | **25.2** | **31.7** | 36.5 |

the per-type breakdown are summarized in Table 3. To better diagnose which spatial skills are most sensitive to viewpoint changes, we further categorize questions into four types based on the reasoning skill required: Distance (nearest/farthest object), Direction (relative spatial orientation), Counting (enumerating objects in a region), and Existence (verifying object presence).

**3D-LLMs struggle with viewpoint rotation.** All models show a steep decline in EM Refined as the number of required correct rotations increases. Even GPT4Scene, the best-performing model, drops from 55.5% at one rotation to merely 0.5% at four rotations, while the remaining models collapse to near zero: LEO (0.4%), 3D-LLM (0.1%), Chat-Scene (0.1%), and Chat-3D v2 (0.0%). Overall VRS scores range from 6.6% to 18.2%, underscoring the models' inability to maintain consistent spatial reasoning across viewpoints. We hypothesize that this arises because existing 3D-LLMs have never been explicitly evaluated on rotation robustness, leading current designs to overlook rotation-invariant feature encoding, cross-view alignment, and fine-grained spatial relationship modeling.

**Rotation failure is architecture-agnostic.** This collapse is universal regardless of whether models adopt point-wise feature encoding (3D-LLM) or object-centric representations (Chat-3D v2, LEO, Chat-Scene, GPT4Scene), indicating a systemic limitation rather than a design-specific issue. Interestingly, 3D-LLM, despite having the second-lowest VRS (9.6%), achieves the highest existence-type score (44.2%), outperforming GPT4Scene (36.5%) and Chat-3D v2 (21.2%). We attribute this to its point-wise encoding, which preserves richer local geometric cues beneficial for object presence detection, whereas object-centric approaches may lose such fine-grained information through feature aggregation. In contrast, distance and direction questions remain consistently challenging across all architectures, confirming that current 3D-LLMs broadly lack viewpoint-invariant spatial representations.

## 5.3 How can we improve 3D-dependency in Real-3DQA?

**Comparison of three training strategies.** We compare the effectiveness of all three training strategies in Tab. 4, evaluated with the EM Refined metric. Initially, we find that the Blind FT models yields notably lower scores for both LEO and Chat-Scene on Real-3DQA. Again, we confirm that Real-3DQA questions are heavily 3D-dependent. However, on our Real-3DQA benchmark, 3D-reweighted fine-tuning demonstrates significant improvements over traditional methods. For LEO, it achieves 29.3 (vs. 19.1 for SFT and 13.6 for Blind FT), while for Chat-Scene, it reaches 33.9 (vs. 22.1 and 14.4). These results confirm that our method successfully encourages models to leverage 3D context for spatial reasoning tasks. To further test generalizability of the finding beyond the SQA3D dataset, we additionally construct another filtered bench-

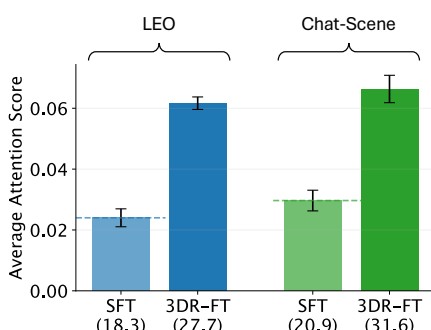

Figure 5: **3D tokens attention score after 3DR-FT.**

mark, *Real-ScanQA*, from the ScanQA test set using the same pipeline described in Sec. 5.2. On *Real-ScanQA*, 3DR-FT improves LEO to 13.9 EM Refine (vs. 6.1 for Supervised FT and 5.9 for Blind FT), mirroring the gains observed in Real-3DQA and highlighting the benefit of 3D-aware training when questions require genuine 3D evidence.

Table 4: **Ablation Study on Training Strategies.** Columns group results by model and dataset: LEO on ScanQA/Real-ScanQA (left) and SQA3D/Real-3DQA (center), and Chat-Scene on SQA3D/Real-3DQA (right). 3D-reweighted fine-tuning (3DR-FT) delivers consistent gains across both *datasets* and *models*, with the largest improvements on the 3D-dependent sets—Real-3DQA and Real-ScanQA—while Supervised FT remains strongest on SQA3D.

| Training Strategy | LEO | | ⟵ LEO ⟶ | | Chat-Scene | |
|---|---|---|---|---|---|---|
| | *ScanQA* | *Real-ScanQA* | *SQA3D* | *Real-3DQA* | *SQA3D* | *Real-3DQA* |
| Supervised FT | **32.3** | 6.1 | **52.2** | 19.1 | **57.2** | 22.1 |
| Blind FT | 33.0 | 5.9 | 50.6 | 13.6 | 51.4 | 14.4 |
| 3D-reweighted FT | 31.3 | **13.9** | 48.2 | **29.3** | 48.9 | **33.9** |

**3D-reweighted finetuning does help 3D-dependency.** As shown in Figure 5, we provide a analysis of attention scores for 3D tokens, following Zhang et al. (2025a), which demonstrates the models' overall dependency on 3D tokens when answering questions. Please refer to the appendix for more implementation details. The results demonstrate that after applying 3D-Reweighted Fine-tuning (3DR-FT), models exhibit significantly higher average attention scores to 3D tokens for both 3D-LLMs. These findings align with the performance gains observed on Real-3DQA. The increase in attention scores confirms that our 3DR-FT strategy not only enhances model performance but, more importantly, successfully guides the models to rely more on 3D visual information rather than textual shortcuts when making spatial reasoning.

**Analysis on Failure Cases.** Table 4 shows a seemingly counterintuitive result: while 3DR-FT substantially improves performance on the 3D-dependent Real-3DQA test set, the overall performance on the original SQA3D test set decreases; for example, Chat-Scene drops from 57.2 to 48.9 after 3DR-FT. At first glance, this seems puzzling—why would emphasizing 3D information hurt performance?

A breakdown of failure cases reveals that this decrease is mainly driven by the 3D-independent portion of SQA3D, as shown in Figure 6. After 3DR-FT, 591 questions flipped from correct to incorrect, and 441 of them were from the filtered set. Notably, nearly 70% (413/591) of these degraded questions were also answered correctly by the text-only (blind) model, suggesting they were shortcut-solvable when using language priors. Many of these questions fall into reasoning types, such as counting (92), navigation (86), visibility (94), and spatial relations (93). We hypothesize that these questions can often be guessed from surface language and answer priors, but become harder when answers must be grounded in genuine 3D input. Thus, the observed drop does not indicate weaker 3D understanding; rather, it mainly reveals how inflated scores on prior benchmarks were driven by linguistic shortcuts, further validating the necessity of our debiased dataset filtering and the proposed VRS metric score.

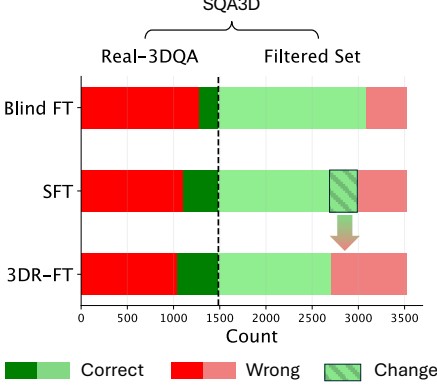

Figure 6: **Why 3DR-FT reduces SQA3D performance?** For Chat-Scene, 591 questions flip from correct to wrong after 3DR-FT; 441 of these come from the Filtered Set (green box with diagonal hatching). Because SQA3D mixes 3D-dependent questions (Real-3DQA) with 3D-independent ones (Filtered Set), emphasizing 3D evidence via 3DR-FT can hurt the latter set, lowering the overall SQA3D score.

## 6 CONCLUSION

This work examines the 3D reasoning capabilities of 3D-LLMs, revealing that state-of-the-art models can rely on textual shortcuts rather than true 3D spatial understanding on existing SQA3D dataset. To address this, we introduce Real-3DQA, a benchmark designed to minimize textual shortcuts and enforce genuine 3D reasoning. Through rigorous filtering and the Viewpoint Rotation Score, our benchmark provides a more robust evaluation framework. Additionally, we propose a 3D-reweighted training strategy that enhances models' reliance on 3D context, significantly improving the spatial

reasoning performance for existing 3D-LLM models. Our findings underscore the need for stronger evaluation benchmarks and training methodologies to advance 3D-LLMs.

# 7 FUTURE WORK

For rotation augmentation, we currently use four common angles to generate 3,000 questions. We believe that considering more rotation angles (such as the 12 o'clock direction) could be valuable. However, this would require more complex rules to define these directions and angles, as well as precise language descriptions to match them, ultimately necessitating re-annotation and significantly increasing manual effort. Nevertheless, from the perspective of expanding rotation robustness, we consider this a promising direction worth exploring. Additionally, investigating model architectures and training schemes that are inherently robust to viewpoint rotation remains an open and equally important direction.

Beyond question answering, we have observed similar issues related to shortcut learning and lack of spatial consistency in 3D grounding and captioning tasks. However, current benchmarks in these areas do not incorporate situated information; all existing datasets adopt an allocentric (scene-centric) perspective rather than an egocentric (observer-centric) one. To enable a meaningful evaluation of spatial understanding in grounding and captioning, future work will require redefining these tasks with egocentric situation descriptions and constructing new QA or caption variants under viewpoint changes. This would involve significant annotation efforts, which we leave as promising future directions to broaden the scope of spatial reasoning evaluation and inspire further attention in the community.

# 8 ACKNOWLEDGEMENTS

We would like to thank Xiongkun Linghu and Jiangyong Huang for generously providing object labeling and detailed scene graph annotations of ScanNet scenes, which were instrumental in the construction of our benchmark. We are grateful to Brandon Smart for carefully proof-reading the manuscript. We also sincerely appreciate Lixiong Chen, Zhenyu Chen, Renrui Zhang, Ian Reid, Guanqi Zhan, Alexei Efros, David Forsyth, Ingmar Posner, and Sijin Chen for their insightful discussions and constructive feedback throughout the development of this work.

# 9 ETHICS STATEMENT

This work adheres to the ICLR Code of Ethics. In this study, no human subjects or animal experimentation was involved. All datasets used, including SQA3D, ScanQA, were sourced in compliance with relevant usage guidelines, ensuring no violation of privacy. We have taken care to avoid any biases or discriminatory outcomes in our research process. No personally identifiable information was used, and no experiments were conducted that could raise privacy or security concerns. We are committed to maintaining transparency and integrity throughout the research process.

# 10 REPRODUCIBILITY STATEMENT

We have made every effort to ensure that the results presented in this paper are reproducible. All code and datasets have been made publicly available in the repository to facilitate replication and verification. We have also provided a full description of dataset construction and reweighted finetuning strategy, to assist others in reproducing our experiments. Additionally, all pretrained models we evaluate, such as 3D-LLM, Chat-3D v2, LEO, Chat-Scene and GPT4Scene, are publicly available, ensuring consistent and reproducible evaluation results. We believe these measures will enable other researchers to reproduce our work and further advance the field.

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

# A    OUTLINE OF THE APPENDIX

- LLM usage(Section B)
- More experiments and analysis
    - Details of how we calculate 3D token attention score (Section C.2)
    - Ablation on LEO Inference (Table 6)
- Details of benchmark construction
    - GPT prompt template for rotation-augmentation (Figure 9)
    - Statistics and examples of filtering 3D-independent questions (Table 7)
    - Data quality control (Section D.2)
- Theoretical insights of 3D-reweighted Fine-tuning (Section E)

# B    LLM USAGE

Large Language Models (LLMs) were used to aid in the writing and polishing of the manuscript. Specifically, we used an LLM to assist in refining the language, improving readability, and ensuring clarity in various sections of the paper. The model helped with tasks such as sentence rephrasing, grammar checking, and enhancing the overall flow of the text.

It is important to note that the LLM was not involved in the ideation, research methodology, or experimental design. All research concepts, ideas, and analyses were developed and conducted by the authors. The contributions of the LLM were solely focused on improving the linguistic quality of the paper, with no involvement in the scientific content or data analysis.

The authors take full responsibility for the content of the manuscript, including any text generated or polished by the LLM. We have ensured that the LLM-generated text adheres to ethical guidelines and does not contribute to plagiarism or scientific misconduct.

# C    MORE EXPERIMENTS AND ANALYSIS

## C.1    PERFORMANCE DROP ACROSS QUESTION TYPES.

To analyze model performance differences across question types, we provide a detailed breakdown in Tab. 5 and a visualization in the radar chart in Fig. 7. First, we observe that all the question types undergo a significant performance drop, further confirming that Real-3DQA's more rigorous nature over SQA3D. Particularly, we observe that most 3D-LLMs consistently struggle with spatial relationships, shape and state recognition, and reasoning tasks. Two of the largest drops in EM Refine appear for the shape and state recognition questions, where five models (3D-LLM/Chat-3D v2/LEO/Chat-Scene/GPT4Scene) drop 55.7/67.8/51.5/47.2/33.3 respectively for shape and 56.5/57.8/49.4/52.7/26.2 respectively on state. Moreover, we find that reasoning questions already had lower scores in SQA3D, particularly for 3D-LLM at 40.0, LEO at 35.0, and Chat-3D v2 at 50.0, despite GPT4Scene achieving 52.5. The multi-hop 3D reasoning is considered the most challenging problem for all current 3D-LLMs. When evaluated on Real-3DQA, these scores plummet further where all five models (3D-LLM/Chat-3D v2/LEO/Chat-Scene/GPT4Scene) drop 40.0/45.4/21.4/21.8/25.2 respectively. Finally, we find that even some basic question types, including color and object identification, become substantially harder in Real-3DQA, with even the best-performing GPT4Scene experiencing a 18.6 point drop in color recognition and a 23.0 point drop in object identification.

Together, these findings reinforce that once 3D-independent and guessable questions are filtered out, current 3D-LLMs routinely fail to demonstrate a robust 3D understanding capabilities and highlight the necessity of the Real-3DQA. We hope our new benchmark will draw the community's attention to the "3D shortcut" issue and help the development of methods that genuinely comprehend 3D context. This multi-dimensional analysis enables more comprehensive comparison of model differences and strengths, highlighting each model's capability preferences and common weaknesses.

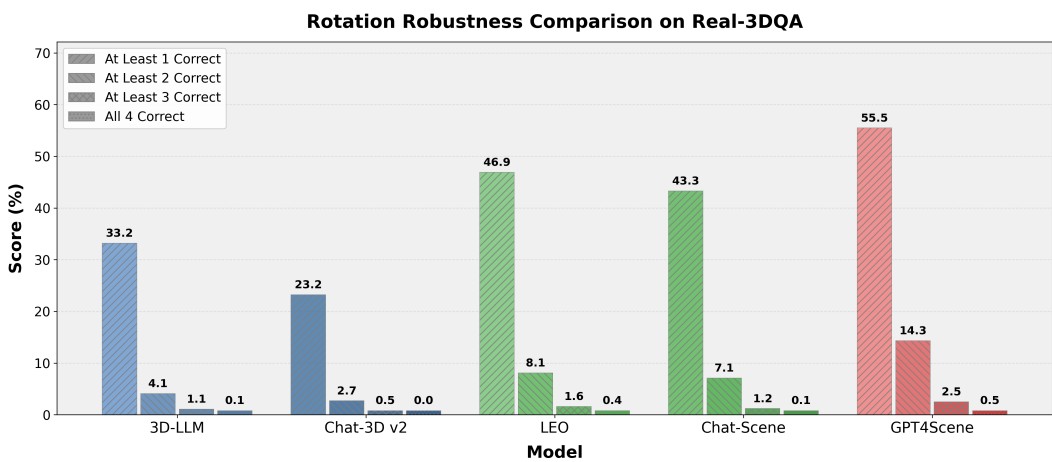

Figure 7: **Comparison of 3D-LLMs capabilities across different question types on Real-3DQA.** The radar chart visualizes the performance of five models (LEO, Chat-Scene, 3D-LLM, Chat-3D v2, GPT4Scene) on ten question categories. Each axis represents a specific question type, and the distance from the center indicates the model's performance in that category. The area covered by each model's polygon reflects its overall capability across all question types. The chart highlights that GPT4Scene demonstrates superior performance in all categories.

Figure 8: **Rotation robustness comparison.** All models show a consistently increasing decline in EM Refined as the viewpoint changes from one to four, underscoring the models' difficulty in maintaining consistency across multiple viewpoints.

Table 5: **Model ability across different question types.** We compare model performance on both the original SQA3D benchmark and our more challenging Real-3DQA. On Real-3DQA, we find most 3D-LLMs have consistent deficiencies in spatial relationships, shape recognition and reasoning tasks. We report refined exact match for detailed question types.

| 3D-LLMs | EM | EM_R | measurement | color | number | spatial relation | shape | state | object | visibility | navigation | reasoning | other |
|---|---|---|---|---|---|---|---|---|---|---|---|---|---|
| | | | | *Refined EM on SQA3D* | | | | | | | | | |
| 3D-LLM | 47.8 | 49.6 | 69.2 | 45.7 | 52.4 | 35.7 | 59.2 | 64.7 | 42.4 | 63.5 | 41.5 | 40.0 | 39.5 |
| Chat-3D v2 | 45.0 | 48.1 | 36.1 | 57.0 | 56.0 | 43.2 | 67.8 | 71.5 | 60.4 | 71.1 | 45.7 | 50.0 | **88.4** |
| LEO | 49.6 | 52.2 | 73.1 | 58.1 | 52.8 | 40.8 | 59.9 | 69.9 | 52.8 | 66.4 | 41.5 | 35.0 | 76.7 |
| Chat-Scene | 54.7 | 57.4 | 80.8 | 58.1 | 57.2 | 46.8 | 66.5 | 76.0 | 59.2 | 70.0 | 48.1 | 40.0 | 81.4 |
| GPT4Scene | **60.6** | **63.3** | **76.9** | **70.8** | **65.8** | **52.6** | **73.7** | **83.7** | **68.8** | **73.3** | 43.3 | **52.5** | 79.1 |
| | | | | *Refined EM on Real-3DQA* | | | | | | | | | |
| 3D-LLM | 7.5 | 10.4 | 0.0 | 10.1 | 9.8 | 9.3 | 3.5 | 8.2 | 10.6 | 4.0 | 18.2 | 0.0 | 20.0 |
| Chat-3D v2 | 3.4 | 9.7 | 20.0 | 5.1 | 13.5 | 9.1 | 0.0 | 13.7 | 3.9 | 17.1 | 12.2 | 4.6 | 0.0 |
| LEO | 14.3 | 19.1 | 0.0 | 29.0 | 20.2 | 16.6 | 8.8 | 20.5 | 22.9 | 11.8 | 17.4 | 13.6 | 60.0 |
| Chat-Scene | 17.0 | 22.1 | **60.0** | 28.3 | 15.2 | 22.3 | 19.3 | 23.3 | 33.0 | 15.8 | 20.6 | 18.2 | 20.0 |
| GPT4Scene | **33.1** | **36.9** | 40.0 | **52.2** | **34.7** | **31.1** | **40.4** | **57.5** | **45.8** | **30.3** | **29.6** | **27.3** | **60.0** |

Table 6: **Where does the 3D prior come from in LEO Huang et al. (2024b) on SQA3D?** We conduct this ablation during only the inference time, using checkpoints provided by the authors. Results show that 3D information and questions are crucial for performance, while situation descriptions have minimal impact. We report both exact match (EM) and refined exact match (EM_Refined) metrics.

| Situation | Question | 3D | EM (%) | EM_R (%) |
|---|---|---|---|---|
| ✓ | ✓ | ✓ | 49.4 | 52.2 |
| ✓ | ✓ | shuffled | 44.2 | 46.7 |
| ✓ | ✓ | ✗ | 32.4 | 43.3 |
| ✗ | ✓ | ✓ | 49.3 | 51.7 |
| ✓ | ✗ | ✓ | 0.2 | 10.4 |
| ✗ | ✓ | ✗ | 18.6 | 30.1 |

## C.2 DETAILS OF 3D TOKEN ATTENTION SCORE CALCULATION.

We randomly selected 100 questions from Real-3DQA, and used both original and 3D-Reweighted fine-tuned versions of LEO and Chat-Scene models for inference. Specifically, we followed the approach in Zhang et al. (2025a), which quantifies models' overall dependency on 3D point cloud tokens when answering questions. By calculating the average attention scores from 3D point cloud tokens to answer tokens, we can determine how much the model relies on 3D information when responding to questions. To ensure statistical significance, we performed three separate sampling runs, collected three sets of data, and calculated standard deviation error bars.

## C.3 ABLATION ON LEO INFERENCE

Table 6 presents an ablation study on LEO to evaluate the impact of different input components on inference performance using the SQA3D benchmark. The study examines how 3D information, situational descriptions, and question relevance influence exact match (EM) and refined exact match (EM_R) scores. It shows by removing 3D input alone, the model can still obtain 32.4% on EM and 43.3% on EM_R, indicating that part of the model prediction is not rely on 3D information. On the other hand, by removing situation description, the model's EM% and EM_R% metrics barely dropped.

## D DETAILS OF BENCHMARK CONSTRUCTION

### D.1 STATISTICS AND EXAMPLES ON FILTERING 3D-INDEPENDENT QUESTIONS

Table 7 shows filtering statistics from SQA3D using our proposed methods in Section 3.1 of the main paper. The table compares the number of filtered and remaining questions across different filtering

methods. Specifically, GPT-4o-mini is used for GPT-based filtering, while LEO, Chat-Scene, and 3D-LLM are employed for SFT and Blind Fine-tuned model comparison.

We find that GPT-based filtering removed only 112 questions, leaving 3407 out of the original 3519. This indicates that GPT filtering alone is relatively conservative, missing many questions that might be easy-to-guess or non-3D-dependent. In contrast, our model comparison approach filters substantially more questions.

Table 7: **Statistics of filtered questions for each filtering step.** The table reports the number of questions filtered and those remaining for the original test set, GPT-based filtering, and model-based comparisons filtering across LEO, Chat-Scene, and 3D-LLM.

| Filter Type | Original Test | GPT | Model Comparison | | |
|---|---|---|---|---|---|
| | | | LEO | Chat-Scene | 3D-LLM |
| **Filtered** | - | 112 | 1197 | 459 | 232 |
| **Remaining** | 3519 | 3407 | 2176 | 1717 | 1485 |

We also show three representative examples of filtered questions to highlights the effectiveness of our filtering approach through model comparison. These filtered questions are so directional that the answer is easily guessable.

- **Question:** What is to my left that gives me natural light in the room?
  **Answer:** window
- **Question:** What is to my right that I can use washable marker to write with?
  **Answer:** whiteboard
- **Question:** What instrument in front of you is ebony and ivory?
  **Answer:** piano

### D.2 QUALITY CONTROL

To ensure the reliability of our rotation-augmented dataset, we adopt a multi-stage quality control pipeline that combines GPT prompting strategies with structured expert review. This process addresses both the risk of hallucination in GPT outputs and the need for consistent, rotation-sensitive annotations.

#### D.2.1 HALLUCINATION MITIGATION

A key concern when leveraging GPT for question augmentation is the potential issue of hallucination. To mitigate this, we employ a two-pronged strategy.

**Instruction-following enhancement via ICL examples.** We designed structured in-context learning (ICL) examples (Figure 9) to guide GPT's generation. These examples reinforced the desired behavior—producing faithful situation texts that simulate specific 3D rotations—and helped GPT adhere to the intended transformation format.

**Manual verification with dual criteria.** We further applied a two-stage manual validation process. Specifically, we retained only those QA pairs that satisfied both: (i) the situation text is rotation-sensitive and accurately reflects the intended 3D rotation; (ii) the original question remains meaningful after rotation, with the updated answer remaining correct in the rotated context. Examples failing either criterion were discarded or corrected. These measures minimize hallucination and ensure that the augmented QA pairs are semantically coherent and grounded in 3D reasoning.

#### D.2.2 EXPERT REVIEW

Beyond hallucination control, we implemented a structured expert review protocol to validate annotation quality.

**Initial filtering of GPT outputs.** Despite careful prompting, about 5% of GPT outputs proved invalid (e.g., incorrect answers after rotation or inconsistent situation descriptions, see Figure 10). To detect

```
system_prompt = {'role':'system','content':"""Your job is to augment the testing data by
rotations for 3D VLM evaluation. The rotation is 90, 180, or 270 degree along the gravity-axis.
You are given a ego-centric scene description for all objects in a room with distances,
orientations, heights from the floor, attributes, and relationships between other objects. You
are also given a situation, a question, and an answer. Your tasks:
1. Read the scene description and understand the spatial relationships between the observer and
the objects in the room.
2. Rotate the situation but keep the question exactly the SAME. DO NOT change the question.
3. Generate the new answer based on the new situation. Make sure the new answer is still valid
based on the scene description.
4a. Output the new situation, question (should be the same as the original), and the new answer.
4b. Try to find a situation that the answer is DIFFERENT from the original answer but still
VALID and explicitly mentioned in the scene description in the correct direction after rotation.
4c. If you cannot find a valid answer under current rotation, output "skip" and do not generate
any new situation.
5. If the situation cannot be rotated meaningfully (no explicit direction), please output
directly "skip" and do not generate any new situation or question.
6. You need to keep the style of the original situation. Keep the length of the answer same as
the original. For most answers, a single word or two is enough. Uncapitalized unless it is an
acronym or proper noun like TV.
7. In scene description, the object are named with a type and a number, e.g., chair-13. You only
refer to the objects by names in situation and question. The number is to distinguish between
objects. Do not leak the number.

[Example Input]
Scene:
  - bed-1: dist=0cm; orient=front; obj_height=43cm; color=white; shape=rectangular;
material=wood; usage=sleeping; texture=smooth; structure=frame with mattress; state=made; on the
top of floor-2;
  - window-2: dist=103cm; orient=right; obj_height=154cm; color=white; material=glass;
usage=lighting; texture=smooth; structure=frame with glass; state=clean; embedded into wall-3;
  - door-3: dist=200cm; orient=left; obj_height=200cm; color=white; material=wood;
usage=entrance; texture=smooth; structure=frame with handle; state=closed; embedded into wall-3;
Situation: I am standing facing the bed in the bedroom.
Question: What can I see on my left?
Answer: door
Rotation: 270

[Example Output]
1. Situation: I am standing facing the window in the bedroom.
2. Question: What can I see on my left?
3. Answer: bed

[Explanation] If I rotate 270 degree (=rotate 90 degree to the right), the bed will be on my
left. The answer should be bed."""}

for angle in [90, 180, 270]:
  messages = [system_prompt]
  messages.append({
    'role':'user',
    'content':f'Scene:\n{scene}\nSituation: {s}\nQuestion: {q}\nAnswer: {a}\nRotation: {angle}'
  })
  output = query(messages)
```

Figure 9: **Prompt Template for Rotation-Augmented Question Generation.** We design this template to guide GPT-4o-mini in creating perspective-rotated variants of the original questions. The template instructs the model to preserve the original question's intent while altering the spatial reference frame according to specified viewpoint changes.

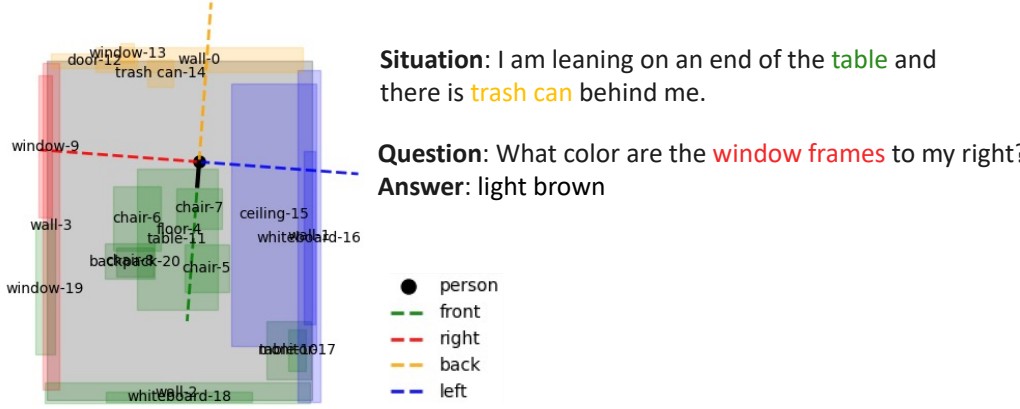

Figure 10: **Example of a view-specific question that cannot be rotated.** This question demonstrates a case where rotation is invalid because it references objects (window frames) that only exist in the current viewpoint (to my right). When the perspective is rotated, these referenced objects are no longer visible or identifiable, rendering the question meaningless in the new viewpoint.

these, we visualized the observer's position and orientation, determined object quadrants (front, back, left, right) from bounding box centers, and checked whether GPT's rotated situations and answers aligned with the new viewpoint. Inaccurate answers were corrected, while fundamentally invalid situations were discarded. Ultimately, we retained 750 question–answer pairs valid across all four viewpoints, yielding $750 \times 4 = 3000$ high-quality pairs.

**Structured protocol.** We further enforced a three-phase expert review protocol: - *Pre-review*: annotators received explicit evaluation criteria and passed a qualification test before large-scale review. - *During review*: experts iteratively tracked errors in a shared log (e.g., mismatched rotations, mishandled states, object counting errors). Random spot checks and annotation speed monitoring ensured consistency. - *Post-review*: annotation reliability was measured with agreement metrics, yielding Cohen's Kappa of 92% (strong inter-rater agreement) and self-consistency of 97%.

Together, these steps guarantee that the final augmented dataset is rotation-sensitive, reliable, and well-suited for benchmarking genuine 3D reasoning.

# E   THEORETICAL INSIGHTS OF 3D-REWEIGHTED FINE-TUNING

To quantify the model's 3D-context dependency, we introduce the *conditional-independence gap* between the ground truth token $\boldsymbol{y}_j$ and the 3D input $\boldsymbol{x}_{3D}$, as

$$\delta_j := \frac{p_\theta(\boldsymbol{y}_j|\boldsymbol{y}_{<j}, \boldsymbol{x}_{\text{text}}, \boldsymbol{x}_{3D}) - p_\theta(\boldsymbol{y}_j|\boldsymbol{y}_{<j}, \boldsymbol{x}_{\text{text}})}{p_\theta(\boldsymbol{y}_j|\boldsymbol{y}_{<j}, \boldsymbol{x}_{\text{text}})}. \tag{3}$$

When $\delta_j = 0$, the token $\boldsymbol{y}_j$ is *conditionally independent* of $\boldsymbol{x}_{3D}$ given the text input $\boldsymbol{x}_{\text{text}}$; in other words, the 3D context is *not* altering the distribution of $\boldsymbol{y}_j$. This is precisely what we aim to avoid in the Real-3DQA benchmark, since it implies that the model is ignoring the 3D information.

Below, we link the conditional-independence gap to our loss function. For simplicity, we denote

$$p_\theta(\boldsymbol{y}_j|\boldsymbol{y}_{<j}, \boldsymbol{x}_{\text{text}}) := s_j, \quad \text{and thus}$$

$$p_\theta(\boldsymbol{y}_j|\boldsymbol{y}_{<j}, \boldsymbol{x}_{\text{text}}, \boldsymbol{x}_{3D}) := s_j(1 + \delta_j). \tag{4}$$

Inserting Eq. (1) in main paper and Eq. (4) into Eq. (2) in main paper leads us to

$$\mathcal{L}_{\text{3DR-FT}}(\theta)$$

$$= \mathbb{E}_{\mathcal{D}}\Big[-\sum_{j=1}^{T} w_j(\boldsymbol{y}, \boldsymbol{x}_{\text{text}})\Big(\log s_j \;+\; \log\big(1+\delta_j\big)\Big)\Big]$$

$$= \underbrace{\mathbb{E}_{\mathcal{D}}\Big[-\sum_{j=1}^{T}\log p_\phi\big(\boldsymbol{y}_j \mid \boldsymbol{y}_{<j}, \boldsymbol{x}_{\text{text}}\big)\Big]}_{\text{Perplexity without 3D input}} \;+$$

$$\underbrace{\mathbb{E}_{\mathcal{D}}\Big[-\sum_{j=1}^{T} w_j(\boldsymbol{y}, \boldsymbol{x}_{\text{text}})\log\big(1+\delta_j\big)\Big]}_{\text{Weighted conditional-independence gap}}.$$

Here, the first term corresponds to the BF model's perplexity on the ground truth. It has zero derivative w.r.t. $\theta$ and, thus, does not affect $\theta$ during training.

Essentially, the 3DR-FT loss function increases $\log(1 + \delta_j)$ during training and leads to a non-zero $\delta_j > 0$ whenever $\boldsymbol{y}_j$ is correctly dependent on the 3D context. Consequently, 3DR-FT pushes the model to rely on 3D information for improved predictions, mitigating the textual shortcut problem observed in standard fine-tuning.

