# OpenReview forum: "Do 3D Large Language Models Really Understand 3D Spatial Relationships?"
_ICLR.cc/2026/Conference — ICLR 2026 Poster_

### Official Review · Reviewer_dfML · 2025-10-31

**Soundness:** 2
**Presentation:** 3
**Contribution:** 2
**Rating:** 6
**Confidence:** 3

**Summary:**

This paper finds that in the current benchmarks for 3D-aware reasoning, as SQA3D, there are many easy-to-guess questions or questions that do not require any 3D input and information. Therefore, the paper introduces Real-3DQA, which is a more rigorous benchmark that filter out the unproper questions for 3D understanding. Furthermore, the paper proposes a 3D-reweighted training objective to guide the model to focus on 3D features when performing 3D reasoning. Experimental results show that the previous high scores in 3D understanding benchmarks largely do not reflect the true 3D reasoning capability of the current 3D-LLMs. The performance on the proposed more rigorous benchmark can more truthfully reflect the real 3D understanding and reasoning capability of 3D-LLMs.

**Strengths:**

++ The paper spotted an interesting yet important issue in the current evaluation of 3D-LLMs, which can raise the awareness of the community on evaluating the true 3D understanding capability of 3D-LLMs.

++ The way of filtering out the 3D-irrelavant questions by comparing the answers of with and without 3D input is reasonable and effective.

++ The writing of this paper is mostly clear. The current issue is clearly stated, followed by the reasonable solutions to build a more reasonable benchmark, and finally shows the performance drop of current method on the more 3D-aware benchmark. Therefore, the overall paper flow is smooth, except for the viewpoint rotation score section which I have some confusions stated below.

**Weaknesses:**

-- Basically, 3D-LLMs are considered to be general purpose AI assistants. Therefore, in real applications, not all questions have to be related to 3D reasoning. In this case, the performance drop on the questions that can be shortcut-solvable by language priors becomes a real issue, because it means that the general usability of the system will get harmed, as there may only be a portion of tasks that truly require 3D reasoning and understanding. Therefore, the proposed 3DR-FT method has flaws that fails to preserve the original reasoning capabilities of LLMs.

-- I do not quite get the meaning and motivation to design such a viewpoint rotation score in Section 3.2. Basically, I think from Section 3.1, the 3D-independent questions have already been filtered out when comparing the answers. Therefore, there seems to be no need to put the effort to create questions that rotate the viewpoints, as creating these questions for all the four viewpoints may need some manual labor.

-- Typo: Line 461: "3D-RFT" should be "3DR-FT".

**Questions:**

-- For the viewpoint rotation score, how are the questions for creating the viewpoint variants getting chosen? From my understanding, it may require heavy manual work, as first we need to select questions that can have different answers when the viewpoints vary. Second, we need to manually label what the answers are for each directions. This is why I think the viewpoint rotation score part feels unnecessary, as it requires manual labor but is not quite justified to be useful.

---

> ### Author Response · Authors · 2025-11-27
> **Author Response to Reviewer dfML (1/3)**
>
> **We sincerely appreciate the reviewer’s feedback and the time dedicated to evaluating our work. We address the concerns as follows.**
>
> >**Basically, 3D-LLMs are considered to be general purpose AI assistants. Therefore, in real applications, not all questions have to be related to 3D reasoning. In this case, the performance drop on the questions that can be shortcut-solvable by language priors becomes a real issue, because it means that the general usability of the system will get harmed, as there may only be a portion of tasks that truly require 3D reasoning and understanding. Therefore, the proposed 3DR-FT method has flaws that fails to preserve the original reasoning capabilities of LLMs.**
>
> **Response:**
> We thank the reviewer for the thoughtful concern about preserving the general-purpose capabilities of 3D-LLMs. We would like to clarify two key points regarding the goals of our benchmark and the design philosophy of 3DR-FT.
>
> **(1) Our benchmark aims to isolate *true* 3D reasoning capability.**
> The purpose of Real-3DQA is not to measure the overall usability of a general AI assistant, but to specifically probe whether a model possesses reliable 3D spatial understanding beyond linguistic shortcuts. Any 3D-LLM, irrespective of its broader language abilities, can be evaluated with this benchmark to assess this particular dimension of capability. Therefore, the benchmark emphasizes 3D reasoning fidelity rather than overall generality.
>
> **(2) 3DR-FT is intended to highlight the 3D-dependency issue rather than serve as a full solution.**
> Our aim with 3DR-FT is not to position it as a universal fine-tuning strategy for all 3D-LLM applications, but rather to help surface and analyze the underlying 3D-dependency problem more clearly.
>
> While 3DR-FT yields measurable gains on Real-3DQA, we recognize that such a lightweight loss change cannot fully resolve the deeper challenge of encouraging models to rely appropriately on 3D information. This limitation itself underscores how challenging and pervasive the 3D-dependency issue is.
> To shed light on these behaviors, we include a detailed error analysis in the main paper, and we hope that making these difficulties explicit will motivate more principled and comprehensive solutions from the community to improve genuine 3D understanding.
>
> In summary, Real-3DQA is intended to faithfully measure a dedicated capability dimension and 3DR-FT should be viewed as a lightweight diagnostic tool that helps reveal, rather than fully solve, the persistent challenge of encouraging 3D-LLMs to depend appropriately on 3D information.

---

> ### Author Response · Authors · 2025-11-27
> **Author Response to Reviewer dfML (2/3)**
>
> >**I do not quite get the meaning and motivation to design such a viewpoint rotation score in Section 3.2. Basically, I think from Section 3.1, the 3D-independent questions have already been filtered out when comparing the answers. Therefore, there seems to be no need to put the effort to create questions that rotate the viewpoints, as creating these questions for all the four viewpoints may need some manual labor.**
>
> **Response:**
> We appreciate the reviewer’s question and apologize for the earlier lack of clarity. At a high level, **both Section 3.1 and Section 3.2 are designed to remove inflated performance exhibited by current 3D-LLMs, so that Real-3DQA can more faithfully evaluate *real* 3D understanding ability**. The two stages address *different sources* of inflation and therefore cannot substitute for each other.
>
> **(1) Section 3.1 removes shortcut-solvable questions, but does not verify true 3D understanding.**
> Blind-model filtering in Section 3.1 is used to remove questions that can be answered purely from linguistic priors. This eliminates one major source of inflated scores—shortcut-driven prediction.
> However, even among the remaining 3D-dependent questions, a model may still answer some items correctly **by random guess rather than through any genuine understanding of the underlying spatial relations**. This leads to a second form of inflated performance: correctness that does not reflect true 3D scene understanding.
>
> Thus, Section 3.1 ensures that the remaining questions *require* 3D cues, but it does **not** guarantee that a model actually attends to those cues and understands them in order to answer correctly.
>
>
> **(2) Section 3.2 evaluates whether the model’s 3D reasoning is geometrically consistent across viewpoints.**
> To eliminate this remaining inflation, Section 3.2 introduces a cross-view consistency test by generating rotated variants. A model that truly understands the 3D layout should answer the *same question* correctly even after the situation is rotated. If a model fails under rotation while the question remains unchanged, the original correctness is likely due to guesswork or view-specific shortcuts rather than genuine spatial understanding.
>
> This is clearly illustrated in **Figure 4 of the main paper**:
>
> - **Original viewpoint:**
>   *Question:* “What is on my right?”
>   *GT:* White board
>   *Model (LEO):* White board ✓
>
> - **After a 90° rotation:**
>   The **trash can** is now on the right.
>   *Question:* “What is on my right?” (unchanged)
>   *GT:* Trash can
>   *Model (LEO):* White board ✗ (simply repeats the old answer)
>
> This demonstrates that the original correct answer did not stem from a viewpoint-aware 3D representation, which Section 3.1 alone cannot reveal.
>
> **In summary**, Section 3.1 removes shortcut-driven inflation, while Section 3.2 removes view-dependent guesswork. **Both steps share the same overarching purpose: to eliminate inflated performance in current 3D-LLMs so that Real-3DQA can measure *real* 3D understanding ability.**
> Only by addressing both sources of inflation can we faithfully evaluate whether a model exhibits genuine, viewpoint-consistent spatial reasoning.

---

> ### Author Response · Authors · 2025-11-27
> **Author Response to Reviewer dfML (3/3)**
>
> >**For the viewpoint rotation score, how are the questions for creating the viewpoint variants getting chosen? From my understanding, it may require heavy manual work, as first we need to select questions that can have different answers when the viewpoints vary. Second, we need to manually label what the answers are for each directions. This is why I think the viewpoint rotation score part feels unnecessary, as it requires manual labor but is not quite justified to be useful.**
>
> **Response:**
> We thank the reviewer for raising this point. The construction of viewpoint-rotated QA variants does *not* rely on heavy manual annotation. The entire pipeline is automated, and the only manual effort is a standard sanity-check step that is routinely required in dataset creation.
>
> **(1) No requirement that answers must change after rotation.**
> We clarify that we do *not* assume that the answer must differ across viewpoints.
> After rotation, the correct answer may stay the same or change, depending solely on the updated spatial configuration. The rotated ground-truth answer is therefore whatever object appears at the correct relational position in the rotated situation—not an artificially enforced change. This resolves a core misunderstanding: we do not pre-select only questions whose answers differ after rotation.
>
> **(2) Automatic generation of all rotated QA pairs.**
> The viewpoint variants—including rotated situation descriptions and the corresponding ground-truth answers—are generated fully automatically.
> Our pipeline takes as input:
>
> - the **original situation description**,
> - the **original question**,
> - the **original ground-truth answer**, and
> - a **structured scene-graph dictionary**, where each object is represented together with its neighboring objects, distances, azimuthal relations, object sizes, and additional attributes such as colors, textures, and functional properties.
>
> This scene-graph dictionary provides a global and explicit encoding of the spatial layout.
> We then prompt an LLM with this structured representation to output the three rotated situations (for 90°, 180°, 270°) together with the *same question* and the *rotated ground-truth answers*.
> The prompt template, including a concrete example of the scene-graph dictionary, is shown in Appendix Fig. 9.
>
> Since the LLM infers new answers directly from the rotated scene graph, **no manual labeling or directional annotation is needed at this stage**.
>
> **(3) Manual checking is lightweight and standard practice.**
> The only manual effort is a final sanity-check over the generated set to ensure correctness. This inspection does *not* involve creating questions or labeling answers; it simply verifies the automatically produced content. Such validation is common practice in dataset construction.
> To make verification efficient and rigorous, we visualize for each question:
>
> - a **top-down view** of the scene,
> - the **observer’s location and facing direction** under the original viewpoint,
> - the **four directional regions** (front/left/back/right), and
> - the **objects contained in each region**.
>
> This visualization (examples shown in **Appendix Fig. 10**) allows annotators to clearly see whether:
>
> 1. the rotated situation description correctly reflects the new viewpoint,
> 2. the rotated answer matches the new geometric configuration, and
> 3. the question remains meaningful after rotation (otherwise it is filtered out).
>
> With this visualization pipeline, a well-trained expert can verify approximately **300 rotated variants per hour**. In total, the full set of **3,000 rotated variants** can be reviewed in roughly **10 hours**, making the required manual effort lightweight and easily manageable.
>
>
> In summary, the viewpoint-rotation pipeline is almost entirely automated, with only minimal human oversight comparable to standard dataset quality control. This enables the Viewpoint-Rotation Score to serve as an efficient and necessary measure of consistent 3D spatial understanding—something that cannot be assessed by Section 3.1 alone.

---

### Official Review · Reviewer_vJtJ · 2025-10-31

**Soundness:** 3
**Presentation:** 3
**Contribution:** 3
**Rating:** 8
**Confidence:** 4

**Summary:**

This paper presents three main contributions to the development and evaluation of 3D large language models (3D-LLMs). 1) it introduces Real-3DQA, a benchmark that filters out questions answerable through textual shortcuts, thereby providing a more rigorous assessment of true 3D reasoning abilities. 2) it proposes the Viewpoint Rotation Score (VRS), a metric that evaluates a model’s spatial consistency under viewpoint changes, reducing the influence of superficial linguistic cues. 3) it develops a 3D-aware Reweighted Fine-Tuning (3DR-FT) method that re-emphasizes 3D-dependent information to enhance spatial understanding. Extensive experiments show that existing 3D-LLMs still exhibit limited real 3D reasoning capabilities on Real-3DQA, while the proposed 3DR-FT significantly improves their 3D comprehension and robustness.

**Strengths:**

- The paper is clearly written and easy to follow, with a well-structured presentation of ideas. The authors’ approach to analyzing the limitations of current 3D-LLMs is insightful and well-motivated, making the paper engaging to read.
- It provides a comprehensive and well-organized overview of existing 3D-LLMs and their evaluation benchmarks.
- The proposed Real-3DQA dataset and VRS metric are meaningful and valuable additions to the 3D-LLM research community. The comparative evaluation of existing 3D-LLMs on both the original and filtered benchmarks is thorough and revealing.
- The ablation studies on the proposed 3DR-FT method are well-designed and conducted across multiple models and datasets. The results are consistent and convincingly demonstrate the effectiveness of the proposed fine-tuning strategy and dataset.

**Weaknesses:**

- The proposed 3D-aware Reweighted Fine-Tuning (3DR-FT) method improves performance on the Real-3DQA benchmark but results in degraded performance on the original datasets. This suggests that the approach may either reduce the model’s generalization to language-prior-heavy questions or bias it toward producing answers that deviate from the dominant linguistic mode when 3D cues are absent. However, in real-world scenarios, many questions that 3D-LLMs encounter are likely to rely heavily on linguistic priors. Therefore, a more detailed discussion on how to balance 3D dependence and language adaptability would strengthen the paper.
- The overall dataset size of Real-3DQA is relatively small, and the variation in scene types, spatial relations, and question templates appears limited. This may restrict the benchmark’s ability to comprehensively evaluate diverse aspects of 3D reasoning and generalization across different environments. The authors are suggested to discuss how to further improve the 3D-LLM evaluation.

**Questions:**

Minor typo: line 226 "Mx. The ..." -> "Mx, the ..."

---

> ### Author Response · Authors · 2025-11-27
> **Author Response to Reviewer vJtJ (1/2)**
>
> **We sincerely appreciate the reviewer’s feedback and the time dedicated to evaluating our work. We address the concerns as follows.**
>
> >**Q1: The proposed 3D-aware Reweighted Fine-Tuning (3DR-FT) method improves performance on the Real-3DQA benchmark but results in degraded performance on the original datasets. This suggests that the approach may either reduce the model’s generalization to language-prior-heavy questions or bias it toward producing answers that deviate from the dominant linguistic mode when 3D cues are absent. However, in real-world scenarios, many questions that 3D-LLMs encounter are likely to rely heavily on linguistic priors. Therefore, a more detailed discussion on how to balance 3D dependence and language adaptability would strengthen the paper.**
>
> **Response:**
> We thank the reviewer for raising this important point. We agree that in real-world scenarios, a 3D-LLM should ideally handle both geometry-intensive questions and language-prior-heavy questions. Below we clarify why the observed performance drop occurs, and how we plan to balance 3D dependence and language adaptability.
>
> **(1) The performance drop does not reflect weakened language reasoning, but reduced reliance on shortcuts.**
> As discussed in the main paper (Failure Case Analysis), the decrease on the original SQA3D benchmark is driven primarily by its **3D-independent portion**.
> After 3DR-FT, 591 questions flipped from correct to incorrect, and **almost 70%** of them (413/591) are questions that the **blind-finetuned model** also answers correctly.
> This strongly suggests that these items are solvable using surface language priors or dataset-specific biases, rather than genuine spatial reasoning.
> Thus, 3DR-FT reduces shortcut behavior rather than harming fundamental linguistic capability.
>
> **(2) 3DR-FT intentionally strengthens the model’s reliance on 3D cues when they matter.**
> This aligns with the goal of Real-3DQA—to isolate true 3D reasoning.
> In Supplement Section E, we further provide a theoretical view in terms of **conditional independence**:
> when the ground-truth answer should depend on 3D input but the model prediction becomes independent of that 3D information, the discrepancy between the text-based and 3D-based predictions acts as a principled signal for reweighting.
> This allows the model to shift learning toward spatial grounding where appropriate.
>
> **(3) Balancing 3D dependence and language adaptability is feasible, and is part of our ongoing work.**
> We fully agree with the reviewer that practical 3D-LLM deployment requires both abilities.
> Fortunately, the 3DR-FT procedure can be incorporated in a **modular and non-destructive manner**, enabling future architectures to maintain language adaptability while improving spatial reasoning.
> Promising directions include:
>
> - **MoE-style routing**: a 3D expert trained with 3DR-FT and a language expert trained conventionally, with query-dependent routing.
> - **Selective reweighting**: activating 3DR-FT only on questions or tokens identified as spatial dependent.
> - **Hybrid training schedules**: alternating training between language-heavy batches and 3D dependent batches.
>
> These strategies can preserve the model’s linguistic flexibility while fully leveraging the benefits of 3DR-FT for spatial grounding.
>
> In summary, 3DR-FT reveals and mitigates shortcut issues rather than diminishing core language ability, and the balance between 3D dependence and language adaptability can be achieved through modular extensions. We appreciate the reviewer’s comment and will highlight this discussion to clarify the practical implications of our method.

---

> ### Author Response · Authors · 2025-11-27
> **Author Response to Reviewer vJtJ (2/2)**
>
> >**Q2: The overall dataset size of Real-3DQA is relatively small, and the variation in scene types, spatial relations, and question templates appears limited. This may restrict the benchmark’s ability to comprehensively evaluate diverse aspects of 3D reasoning and generalization across different environments. The authors are suggested to discuss how to further improve the 3D-LLM evaluation.**
>
> **Response:**
> We appreciate the reviewer’s thoughtful comment regarding dataset scale and diversity. We agree that further expansion in scene types, spatial relations, and question templates would strengthen the benchmark, and we view Real-3DQA as an initial step toward a more comprehensive evaluation suite for 3D-LLMs. Below we outline our planned extensions that directly address the reviewer’s suggestions.
>
> ### Future Plans
>
> **(1) Expanding to MSQA [1] for broader scene and relation coverage**
> Our construction framework is generalizable across datasets.
> In addition to ScanQA [2] and SQA3D [3], we are extending Real-3DQA to **MSQA**, whose scenes combine three major sources: ScanNet (67 scenes, already included in our current version), 3RScan (45 scenes), and ARKitScenes (48 scenes).
> This will expand our benchmark to **160 scenes**, covering more varied spatial layouts and relation types, thereby mitigating concerns about scene diversity.
>
> **(2) Scaling further via ScanNet++ [4] and HM3D [5]**
> To substantially increase scale and environment variety, we plan to integrate
> - **ScanNet++** (1,006 high-fidelity indoor scans), and
> - **HM3D** (1,000+ multi-room, photorealistic scans).
> These datasets offer richer structural diversity—multi-room layouts, long hallways, cluttered environments, and varied furniture arrangements—allowing us to evaluate generalization across a wider spectrum of 3D settings.
>
> **(3) Enriching spatial relations and question templates via finer-grained viewpoint rotations**
> Beyond adding more scenes, we aim to increase relational and linguistic diversity by incorporating **non-orthogonal viewpoint rotations** (e.g., 30°, 60°, 150°, etc.).
> This enables more nuanced spatial queries such as:
>
> > *“What is behind the sofa in my 2 o’clock direction?”*
>
> In this way, a rotation-aware question will no longer be limited to just 3 variants after augmentation, but could potentially have 8 or more meaningful variants that capture richer spatial relations and more diverse reasoning patterns. This will significantly increase the number of valid question variants per scene, greatly enriching both the scale and difficulty of the benchmark.
>
> To sum up, these expansions will significantly enhance the coverage of environments, spatial configurations, and linguistic structures in Real-3DQA. We hope this clarifies our long-term vision and demonstrates our commitment to developing a more comprehensive benchmark for evaluating diverse aspects of genuine 3D reasoning in 3D-LLMs.
>
> ---
>
> **References**:
>
> \[1] MSQA: Multi-modal Situated Reasoning in 3D Scenes
>
> \[2] ScanQA: 3D Question Answering for Spatial Scene Understanding
>
> \[3] SQA3D: Situated Question Answering in 3D Scenes
>
> \[4] Scannet++: A high-fidelity dataset of 3d indoor scenes
>
> \[5] Habitat-Matterport 3D Dataset (HM3D): 1000 Large-scale 3D Environments for Embodied AI

---

### Official Review · Reviewer_SPo3 · 2025-10-31

**Soundness:** 3
**Presentation:** 3
**Contribution:** 4
**Rating:** 8
**Confidence:** 4

**Summary:**

The paper asks whether current 3D-LLMs truly understand 3D spatial relationships. The authors first show that a **text-only “blind” fine-tuned LLM** can match or surpass several 3D-LLMs on SQA3D, implying the benchmark allows shortcutting via linguistic priors rather than genuine 3D reasoning. They therefore construct Real-3DQA, filtering out questions that both a vision-conditioned model and its blind counterpart (and then a general LLM) can answer correctly without 3D input, yielding a set more dependent on 3D evidence. They further introduce a rotation-robustness metric, the Viewpoint Rotation Score (VRS), which evaluates consistency across rephrased, viewpoint-rotated variants of the same question. Experiments reveal sharp drops in accuracy when requiring correctness across all rotated views, highlighting poor rotation consistency in recent 3D-LLMs. Finally, the paper proposes 3D-aware Reweighted Fine-Tuning (3DR-FT), which uses a blind-model reference to upweight samples that are hard to guess from text alone, substantially improving performance on Real-3DQA (and a similarly filtered Real-ScanQA) while encouraging reliance on 3D cues.

**Strengths:**

The paper is original in framing “real” 3D understanding via the Real-3DQA pipeline and the rotation-consistency metric (VRS), moving beyond shortcut-prone benchmarks. Methodologically it’s solid: the blind-vs-vision contrast, rotated rephrasings, and multi-stage QC make the evidence credible, and 3DR-FT is a clear, effective objective. The writing and figures communicate the pipeline and metrics cleanly. Empirically, the work exposes meaningful brittleness in 3D-LLMs and shows a practical path (3DR-FT) to increase true 3D reliance—making the contribution significant.

**Weaknesses:**

Rotation robustness is evaluated using GPT-generated viewpoint texts. Although quality control is thorough, this approach still relies on linguistic rather than geometric variation and limits the test’s realism. Using scene-graph or pose-based rotations with automatic answer recomputation would more directly assess spatial consistency. Coverage is limited to four fixed views; expanding to denser yaw and pitch/roll, plus reporting uncertainty would strengthen claims. The Real-3DQA set is much smaller post-filtering, so testing robustness to alternative filtering unions would address potential distributional shifts. 3DR-FT is demonstrated on two backbones and trades off SQA3D performance; broader ablations and a multi-objective/curriculum variant could maintain gains on Real-3DQA without degrading shortcut-heavy sets.

**Questions:**

1. How sensitive is Real-3DQA to the choice of “blind” baseline? Would using different text-only LLMs (e.g., smaller or instruction-tuned ones) change which questions are filtered out?
2. Could the authors validate that the GPT-generated rotated descriptions correspond to actual geometric viewpoint shifts rather than linguistic paraphrases? A small human-verified subset or rendered 3D examples would help.
3. What are the main failure modes of 3DR-FT—does it hurt generalization to unseen object types or tasks beyond QA?
4. The four fixed rotation angles cover only azimuth changes. Would adding elevation or roll perturbations further reduce accuracy?

---

> ### Author Response · Authors · 2025-11-27
> **Author Response to Reviewer SPo3 (1/3)**
>
> **We sincerely appreciate the reviewer’s feedback and the time dedicated to evaluating our work. We address the concerns as follows.**
>
> >**W1: Rotation robustness is evaluated using GPT-generated viewpoint texts. Using scene-graph or pose-based rotations with automatic answer recomputation would more directly assess spatial consistency.**
>
> **Response:**
> We thank the reviewer for the thoughtful suggestion. We did consider, and even experimented with, using scene-graph or pose-based rotation combined with automatic answer recomputation. While this approach appears more “geometric” in principle, in practice it introduces several challenges that make it less reliable and more labor-intensive than expected.
>
> First, directly rotating situations and recomputing answers using geometric rules requires strict, carefully designed heuristics. For example, when the camera rotates by 90°, one must decide: What if no object exists in the new target direction? What if **multiple** objects occupy that region? Which one should be selected as the answer, and is that object semantically meaningful for the question (e.g., walls, ceilings, large structures)? Handling these cases robustly requires a large number of handcrafted rules, each potentially introducing new edge cases.
>
> Second, updating the textual situation description in a purely geometric manner tends to yield **unnatural or rigid language**, especially for complex indoor scenes. Ensuring that the updated text remains natural, coherent, and aligned with what a human would expect requires substantial manual correction, reducing the benefit of the fully-automatic approach.
>
> Third, despite the effort required to define rotation rules, **manual verification remains unavoidable**. Ambiguous spatial configurations, occlusion cases, and complex multi-object arrangements still require human checking to ensure correctness. In our early attempts, we found that the rule-based pipeline increased rather than reduced the amount of manual effort needed.
>
> For these practical reasons, we ultimately employed a GPT-based generation pipeline **conditioned on a structured scene-graph dictionary**, which explicitly encodes spatial relations, distances, azimuthal directions, object sizes, and attributes. This allows the LLM to naturally generate rotated situations and answers while remaining grounded in the underlying geometry. We then apply a lightweight but rigorous human verification step, as detailed in Appendix Fig. 10, to ensure the final results faithfully reflect geometric viewpoint changes.
>
> In summary, while purely geometric rotation and rule-based answer recomputation is conceptually appealing, it requires extensive heuristics and still does not eliminate the need for human checking. Our hybrid scene-graph–guided approach strikes a more practical balance between geometric grounding, linguistic naturalness, and verification cost.
>
> >**W2:Coverage is limited to four fixed views; expanding to denser yaw and pitch/roll, plus reporting uncertainty would strengthen claims.**
>
> **Response:** We appreciate the reviewer’s suggestion. Extending the viewpoint space to denser yaw sampling and incorporating pitch/roll perturbations would indeed provide a more comprehensive robustness evaluation. Our current focus was on azimuthal changes to align with existing datasets and human-viewing conventions, but we agree that richer viewpoint variations and uncertainty estimates would strengthen the benchmark. We consider this a promising direction for future work.
>
> >**W3:The Real-3DQA set is much smaller post-filtering, so testing robustness to alternative filtering unions would address potential distributional shifts.**
>
> **Response:**  We thank the reviewer for raising this point. To address concerns about potential distributional shifts after filtering, we conducted a sensitivity study by varying the choice of blind text-only models (details in our response to Q1). The results show very high overlap across models of different sizes and instruction-following abilities (95–100%), indicating that the filtering process is stable and that Real-3DQA is robust to alternative filtering unions.
>
> >**W4:3DR-FT is demonstrated on two backbones and trades off SQA3D performance; broader ablations and a multi-objective/curriculum variant could maintain gains on Real-3DQA without degrading shortcut-heavy sets.**
>
> **Response:**  We agree with the reviewer that applying 3DR-FT to a broader set of backbones or combining it with multi-objective or curriculum-based training strategies could help retain performance on shortcut-heavy datasets while improving Real-3DQA. Our goal in this work was to show the effect of a simple, diagnostic reweighting scheme on two representative backbones, but we see significant value in exploring richer training pipelines. We view this as an important avenue for future research and plan to investigate these extensions in follow-up work.

---

> ### Author Response · Authors · 2025-11-27
> **Author Response to Reviewer SPo3 (2/3)**
>
> >**Q1: How sensitive is Real-3DQA to the choice of “blind” baseline? Would using different text-only LLMs (e.g., smaller or instruction-tuned ones) change which questions are filtered out?**
>
> **A1:** We thank the reviewer for this insightful question. To evaluate how sensitive Real-3DQA is to the choice of the “blind” baseline, we conducted additional experiments using LEO with different text-only LLMs that vary in both size and instruction tuning status.
>
> Our original blind baseline is **Vicuna-7B**, which is an instruction-tuned version of LLaMA-7B.
> We additionally tested:
>
> - **LLaMA-7B base model** (non–instruction-tuned)
> - **Vicuna-13B** (larger instruction-tuned model)
>
> We compared how many questions each model filters out and computed the overlap with the original Vicuna-7B baseline. The results are:
>
> | Blind Model        | # Filtered | Intersection w/ V-7B | Overlap Rate |
> |--------------------|------------|------------------------|--------------|
> | Vicuna-7B          | 1197       | 1197                   | **100%**     |
> | LLaMA-7B (base)    | 1178       | 1145                   | **95.6%**    |
> | Vicuna-13B         | 1209       | 1181                   | **98.7%**    |
>
> These results show that the filtered question set is **highly stable** across different text-only baselines.
> Model size (7B → 13B) has almost no effect, and even removing instruction tuning (LLaMA-7B base) still preserves over **95%** overlap.
>
> The small decrease for LLaMA-7B base appears reasonable: without instruction tuning, the model has weaker instruction-following and weaker ability to exploit linguistic priors, causing it to identify slightly fewer shortcut-solvable questions. Importantly, the overall differences remain very small.
>
> **In summary, Real-3DQA is not sensitive to the choice of blind model.**  Different text-only LLMs yield nearly identical filtered sets, demonstrating the robustness of our model-based filtering procedure.
>
>
> ---
>
> >**Q2: Could the authors validate that the GPT-generated rotated descriptions correspond to actual geometric viewpoint shifts rather than linguistic paraphrases? A small human-verified subset or rendered 3D examples would help.**
>
> **A2:** We thank the reviewer for the thoughtful question. We believe the concern arises from a misunderstanding: the GPT-generated viewpoint variants in Real-3DQA are *not* linguistic paraphrases. Instead, they are grounded in an explicit structured scene representation that allows GPT to perform genuine geometric viewpoint transformation rather than surface-level rewriting.
>
> **(1) Viewpoint rotation is generated from structured 3D-aware inputs, not free-form text.**
> When generating the rotated variants, GPT is provided not only with the original situation, question, and answer, but also with a **structured scene-graph dictionary** that explicitly encodes the spatial layout:
>
> - the **original situation description**,
> - the **original question** (kept identical across rotations),
> - the **original ground-truth answer**, and
> - a **structured scene graph** describing each object’s neighbors, relative distances, azimuthal relations, sizes, and additional attributes (color, texture, functional properties).
>
> This scene-graph dictionary provides GPT with a global and unambiguous spatial context. Based on this structured representation, the model generates rotated situations for 90°, 180°, and 270° viewpoints along with the corresponding **rotated ground-truth answers**.
> The complete prompt template—including a concrete example of the scene-graph dictionary—is provided in **Appendix Fig. 9**.
>
> Thus, the viewpoint-rotated descriptions are derived from explicit geometric information, not from unconstrained paraphrasing.
>
> **(2) Human verification further ensures that rotated descriptions align with actual geometric viewpoint shifts.**
> We also perform a **full human sanity check** as described in **Section D.2.2 (Expert Review)**.
> To make verification efficient and rigorous, we visualize for each question:
>
> - a **top-down view** of the scene,
> - the **observer’s location and facing direction** under the original viewpoint,
> - the **four directional regions** (front/left/back/right), and
> - the **objects contained in each region**.
>
> This visualization (examples shown in **Appendix Fig. 10**) allows annotators to clearly see whether:
>
> 1. the rotated situation description correctly reflects the new viewpoint,
> 2. the rotated answer matches the new geometric configuration, and
> 3. the question remains meaningful after rotation (otherwise it is filtered out).
>
> Thus, all retained GPT-generated rotations truly correspond to **valid geometric viewpoint transformations**.
>
> In summary, the GPT-generated rotated descriptions are strictly geometry-driven through structured scene-graph inputs, and we further ensure correctness through human verification. This guarantees that the rotated variants reflect true viewpoint shifts rather than linguistic paraphrases.

---

> ### Author Response · Authors · 2025-11-27
> **Author Response to Reviewer SPo3 (3/3)**
>
> >**Q3: What are the main failure modes of 3DR-FT—does it hurt generalization to unseen object types or tasks beyond QA?**
>
> **A3:** We thank the reviewer for the question. To better understand the limitations of 3DR-FT, we analyzed the 591 cases where performance degraded after reweighting. We found that the most frequent failure modes fall into four categories: **counting (92), navigation (86), visibility (94), and spatial relations (93)**. These categories all require *fine-grained and precise* 3D understanding, suggesting that 3DR-FT primarily struggles on questions that demand high-resolution spatial reasoning.
>
> This pattern indicates that, while 3DR-FT successfully encourages models to rely more on 3D cues, it does not by itself grant the model the deeper geometric reasoning abilities needed for these harder question types. In other words, the method helps reduce shortcut behavior but does not fully resolve the underlying challenge of representing and reasoning over complex 3D structure.
>
> We view this as an important insight: 3DR-FT reveals where current 3D-LLMs remain fundamentally limited, and highlights that improving 3D-grounded reasoning—especially for tasks requiring detailed spatial modeling—remains an open and promising direction for future research.
>
> ---
>
> >**Q4: The four fixed rotation angles cover only azimuth changes. Would adding elevation or roll perturbations further reduce accuracy?**
>
> **A4:** We thank the reviewer for the insightful comment. Our rotation perturbations were limited to azimuthal changes because the 3DQA setting is designed to simulate human-like viewing behavior—users typically look around horizontally rather than tilting their viewpoint upward toward the ceiling or downward toward the floor. Moreover, existing 3DQA and SQA datasets (e.g., ScanQA [1], SQA3D [2]) are constructed from scans where the captured viewpoints are almost exclusively azimuth-based. We followed this convention to ensure fair and consistent evaluation across models.
>
> We agree that adding elevation or roll perturbations could further challenge the model and provide complementary insights into 3D robustness. Current datasets such as ScanNet primarily contain single-floor indoor scans, which limits the opportunities for meaningful elevation changes. As 3D benchmarks expand to multi-level environments, this will naturally become feasible. In particular, large-scale datasets such as **HM3D** [3], which contain multi-floor, multi-room reconstructed buildings, provide promising sources for extending Real-3DQA toward vertical reasoning, multi-level navigation, and more diverse viewpoint transformations. We consider incorporating elevation- and roll-based perturbations on such datasets a valuable direction for future work.
>
> ---
>
> **References**:
>
> \[1] ScanQA: 3D Question Answering for Spatial Scene Understanding
>
> \[2] SQA3D: Situated Question Answering in 3D Scenes
>
> \[3] Habitat-Matterport 3D Dataset (HM3D): 1000 Large-scale 3D Environments for Embodied AI

---

### Official Review · Reviewer_L57o · 2025-11-02

**Soundness:** 3
**Presentation:** 3
**Contribution:** 2
**Rating:** 6
**Confidence:** 4

**Summary:**

This paper investigates whether current 3D large language models (3D-LLMs) truly understand 3D spatial relationships, rather than exploiting language priors. The authors show that text-only finetuned LLMs can perform competitively on existing 3D QA benchmarks, suggesting strong linguistic bias. To address this, they propose Real-3DQA, a refined benchmark that filters out 3D-independent questions and introduces viewpoint-rotation consistency evaluation. Additionally, they introduce a simple fine-tuning strategy, 3D-Reweighted Fine-tuning (3DR-FT), which assigns higher weights to questions that are less answerable from text alone. Experiments demonstrate significant performance drops of existing 3D-LLMs on Real-3DQA, confirming the issue, and modest gains from 3DR-FT.

**Strengths:**

- The finding that text-only models perform nearly as well as 3D-LLMs on standard benchmarks is valuable to the community.

- Real-3DQA improves evaluation fairness by filtering 3D-independent samples and introducing rotation consistency, which is sound.

- The authors evaluate multiple existing 3D-LLMs and provide both quantitative and qualitative analyses that convincingly demonstrate the identified problem.

**Weaknesses:**

Limited novelty in methodology: The paper diagnoses dataset bias effectively but does not propose new modeling architectures or mechanisms to fundamentally improve 3D reasoning.
1. The Real-3DQA benchmark largely refines existing datasets via filtering and simple viewpoint augmentations; while useful, it feels more like an engineering refinement than a conceptual leap.
2. The 3DR-FT method adds a weighting term to the loss based on text–3D prediction discrepancy; the idea is intuitive but technically minor.

The work’s value is primarily diagnostic rather than methodological.

**Questions:**

See Weaknesses

---

> ### Author Response · Authors · 2025-11-27
> **Author Response to Reviewer L57o (1/2)**
>
> **We sincerely appreciate the reviewer’s feedback and the time dedicated to evaluating our work. We address the concerns as follows.**
>
> >**The Real-3DQA benchmark largely refines existing datasets via filtering and simple viewpoint augmentations; while useful, it feels more like an engineering refinement than a conceptual leap.**
>
> **Response:**
> We appreciate the reviewer’s perspective and the opportunity to clarify the conceptual motivations behind Real-3DQA. While our benchmark necessarily involves practical filtering and viewpoint augmentation, its design is rooted in a deeper conceptual analysis of why current 3D-QA evaluations fall short.
>
>
> **(1) Why we build on existing datasets rather than creating a new benchmark?**
> We intentionally construct Real-3DQA on top of established 3D-QA datasets instead of introducing an entirely new benchmark from scratch. Doing so allows us to isolate the effect of our 3D-aware filtering pipeline. If we were to simultaneously change the underlying data source and introduce 3D-dependent filtering, it would become difficult to disentangle whether performance differences stem from the 3D-aware filtering or merely from shifts in data distribution. By applying a principled 3D-filtering and viewpoint-consistency augmentation to well-known datasets, we can more cleanly reveal the various 3D-dependency limitations of existing benchmarks.
>
> **(2) Identifying an insightful problem and revealing its root cause can be as valuable as proposing a new architecture.**
> We fully understand the reviewer’s concern regarding the methodological novelty. Our goal in this work, however, is not to introduce another architectural extension, but to provide a clearer understanding of a fundamental evaluation issue in current 3D-LLMs.
>
> Real-3DQA is motivated by the empirical finding shown in **Fig. 2**: a language model fine-tuned only on text QA pairs can match or even outperform several state-of-the-art 3D-LLMs on widely used 3D-QA benchmarks. This observation indicates that existing benchmarks may not adequately measure true 3D reasoning and may unintentionally reward models for leveraging linguistic shortcuts rather than spatial understanding.
>
> We believe that identifying this limitation and analyzing why it arises offers meaningful conceptual value for the community. By making the underlying issue explicit, our work provides a clearer direction for future research. We hope that this benchmark serves as an initial step that encourages more targeted, principled, and robust approaches to evaluating genuine 3D understanding in future 3D-LLMs.
>
> **(3) Real-3DQA introduces a principled, model-based filtering approach.**
> Instead of relying solely on rule-based heuristics or answer-distribution balancing (e.g., SQA3D [1]), we compare each 3D-LLM with its **Blind-FT variant** to identify items solvable without 3D information. This model-level filtering detects deeper annotation artifacts, such as saliency bias, canonical layouts, or inherently guessable questions that purely rule-based methods cannot reliably remove. To our knowledge, this form of consistency-based filtering is relatively uncommon in 3D-LLM evaluation and provides a more robust way to isolate 3D-dependent reasoning.
>
> In addition, we conducted a sensitivity analysis on the choice of blind models, as discussed in the response to **Reviewer SPo3 (2/3 Q1)** . By testing multiple text-only LLMs of different sizes and instruction-following capabilities, we found that the overlap in filtered questions remained consistently high (95–100%). This further demonstrates that the model-based filtering procedure is **stable and robust**, and that our findings are not sensitive to the particular blind baseline used.
>
> **(4) The viewpoint-rotation consistency test adds an additional conceptual dimension.**
> Beyond filtering questions that do not require 3D information, we further examine whether a model maintains *coherent* spatial reasoning across viewpoint changes. This addresses a viewpoint-consistent 3D understanding that current benchmarks do not evaluate, and that cannot be captured through filtering alone. Besides, we find the current models are all very weak on this new dimension.
>
> Taken together, Real-3DQA is motivated not only by engineering considerations but by a conceptual effort to diagnose fundamental weaknesses in existing 3D-QA benchmarks and to introduce principled mechanisms for isolating and assessing genuine 3D reasoning.
>
> **References**:
>
> [1] SQA3D: Situated Question Answering in 3D Scenes

---

> ### Author Response · Authors · 2025-11-27
> **Author Response to Reviewer L57o (2/2)**
>
> >**The 3DR-FT method adds a weighting term to the loss based on text–3D prediction discrepancy; the idea is intuitive but technically minor.**
>
>
> **Response:**
> We thank the reviewer for the comment and appreciate the observation that the idea behind 3DR-FT is intuitive. We agree that the technique is intentionally simple, and we view its effectiveness despite its simplicity as a positive feature. Our intention is not to present 3DR-FT as a comprehensive fine-tuning strategy, but rather as a lightweight mechanism designed to analyze the underlying 3D-dependency issue more clearly.
>
> To clarify the methodological contribution, we provide additional theoretical insights in **Section E of the Supplementary Material**. In particular, 3DR-FT can be interpreted through the lens of **conditional independence**: when the ground-truth answer should depend on the 3D input, but the model’s prediction is effectively conditionally independent of that 3D information (i.e., driven predominantly by textual priors), the discrepancy between the text-based and the 3D-based predictions serves as a signal for reweighting. Under this view, the reweighting term adaptively increases the learning pressure on examples where this conditional dependence is violated, encouraging the model to recover the appropriate reliance on 3D cues rather than linguistic shortcuts. This provides a principled explanation for why such a simple mechanism can mitigate the shortcut behavior commonly observed in standard fine-tuning.
>
> Finally, we also understand the reviewer’s point that “the work’s value is primarily diagnostic rather than methodological.” Indeed, this is aligned with our intention: 3DR-FT is meant to highlight and better expose the pervasiveness of the 3D-dependency problem rather than to claim a complete solution. Supported by the empirical observations in **Fig. 2**, we hope that making these issues explicit will help motivate more targeted and principled methods for strengthening genuine 3D understanding in future 3D-LLM research.

---

### Author Response · Authors · 2025-11-27
**Summary for the New Area Chair**

We would like to express our particular appreciation to the **newly assigned AC**, given this year's unusual situation at ICLR. We fully understand that stepping into an already active review process significantly increases your workload. Our submission receives an overall positive average score (**7**), with a mix of strong recommendations and more reserved assessments (**8, 8, 6, 6**). To help efficiently assess our work, we summarize below the key strengths highlighted by reviewers and the concerns we have addressed during the rebuttal.

---
Across the reviews, several strengths were consistently highlighted:

1. **Identifying a fundamental evaluation gap:**
   Reviewers appreciated our key finding that text-only models can match or even surpass current 3D-LLMs on standard benchmarks, revealing that existing datasets often fail to reflect genuine 3D reasoning (**`Wfm7`**, **`SPo3`**, **`dfML`**).

2. **Conceptual advances in benchmark design:**
   The Real-3DQA benchmark and the Viewpoint Rotation Score (VRS) were seen as meaningful steps beyond shortcut-prone evaluations, with the blind-vs-vision contrast and rotation consistency highlighted as well-motivated contributions (**`Wfm7`**, **`SPo3`**, **`vJtJ`**).

3. **Methodological soundness and robustness:**
   Reviewers noted that our multi-stage QC pipeline, rotated rephrasings, and model-based filtering form a solid methodology, and that the empirical evidence supporting our claims is credible (**`SPo3`**, **`vJtJ`**).

4. **Clarity and organization:**
   Several reviewers found the writing, figures, and overall structure clear and easy to follow, making the motivation and contributions transparent (**`vJtJ`**, **`dfML`**).

5. **Comprehensive empirical analysis and practical insights:**
   Reviewers recognized the breadth of our evaluation—including cross-benchmark comparisons, ablations, and the demonstration that 3DR-FT encourages true 3D reliance—offering both diagnostic insights and practical value (**`Wfm7`**, **`SPo3`**, **`vJtJ`**, **`dfML`**).

---

During the rebuttal, the two reviewers who gave a score of 6 mainly raised clarification-related concerns—particularly regarding the motivation for viewpoint rotation, the details of the rotation-generation pipeline, and the interpretation of 3DR-FT. We have addressed **all** of these through detailed explanations and additional analyses, which are summarized below:


1. **Motivation for viewpoint rotation and the two-stage filtering pipeline:**
   We clarified that Section 3.1 and Section 3.2 address *different* forms of inflated performance (shortcut-driven vs. view-specific guesswork). Both steps are necessary to measure true 3D reasoning.

2. **Scene-graph–grounded rotation generation (not paraphrasing):**
   We explained that rotated situations are generated from a structured scene-graph dictionary rather than through linguistic paraphrasing. A rigorous expert review step (Appendix Figs. 9–10) ensures the rotated descriptions correspond to real geometric viewpoint shifts.

3. **Robustness of model-based filtering:**
   To address concerns about potential distributional shifts, we conducted a **sensitivity study** across multiple blind baselines (Vicuna-7B, LLaMA-7B, Vicuna-13B), showing 95–100% overlap in filtered items, demonstrating high robustness (SPo3 Q1).

4. **Clarification of 3DR-FT’s intent and limitations:**
   We emphasized that 3DR-FT is a lightweight diagnostic tool designed to expose 3D-dependency issues, not a universal fine-tuning strategy. Its limitations further highlight the pervasiveness of shortcut reliance and motivate more principled future methods.

5. **Future-scale extensions:**
   We acknowledged reviewer suggestions regarding denser yaw sampling, pitch/roll perturbations, and broader data diversity. We plan to extend Real-3DQA to multi-floor datasets such as HM3D and incorporate richer viewpoint variations in future iterations.

---

We once again thank all reviewers and especially the new AC for their constructive feedback and substantial time commitment under tight deadlines. We hope that the additional clarifications and analyses help strengthen the presentation and make the conceptual and practical contributions of Real-3DQA and 3DR-FT clearer to the community.

---

### Meta-Review · Area_Chair_Z8gw · 2026-01-07

**Summary:**

This paper received scores of 6, 8, 8, 6 in the initial review. The contributions of such a diagnostic work for spatial understanding and reasoning are commonly acknowledged by all the reviewers. The main concerns are around minor issues for clarification regarding the motivation for viewpoint rotation, the details of the rotation-generation pipeline, and limitations of 3DR-FT as well as the new benchmark. Most of the details are well clarified in the rebuttal, and the limitations are relatively minor compared to the overall contribution. Therefore, the paper is recommended for acceptance.

**Reviewer Concerns:**

Common concerns are focused on the limited novelty and limitations of 3DR-FT and the new benchmark (L57o, SPo3, vJtJ, dfML). The other concerns are related to clarifications regarding the motivation of viewpoint rotation and the details of rotation-generation pipelines, which are most well addressed in the rebuttal. Although the common concerns are hardly fully addressed shortly, reviewers with such a concern weigh more contributions than these minor issues and recommend clear acceptance.

**Reviewer Scores:**

Two reviewers with scores of 6 may consider keeping or raising the scores with the minor issues clarified and the other reviewers are expected to keep the scores of 8. The final decision should have an average acceptance score with a clear agreement.

---

### Decision · Program_Chairs · 2026-01-26

Accept (Poster)